# Integrative modeling reveals the molecular architecture of the intraflagellar transport A (IFT-A) complex

Caitlyn L McCafferty[1]*, Ophelia Papoulas[1], Mareike A Jordan[2], Gabriel Hoogerbrugge[1], Candice Nichols[1], Gaia Pigino[3], David W Taylor[1], John B Wallingford[1], Edward M Marcotte[1]*

[1]Department of Molecular Biosciences, Center for Systems and Synthetic Biology, University of Texas, Austin, United States; [2]Max Planck Institute of Molecular Cell Biology and Genetics, Dresden, Germany; [3]Human Technopole, Milan, Italy

**Abstract** Intraflagellar transport (IFT) is a conserved process of cargo transport in cilia that is essential for development and homeostasis in organisms ranging from algae to vertebrates. In humans, variants in genes encoding subunits of the cargo-adapting IFT-A and IFT-B protein complexes are a common cause of genetic diseases known as ciliopathies. While recent progress has been made in determining the atomic structure of IFT-B, little is known of the structural biology of IFT-A. Here, we combined chemical cross-linking mass spectrometry and cryo-electron tomography with AlphaFold2-based prediction of both protein structures and interaction interfaces to model the overall architecture of the monomeric six-subunit IFT-A complex, as well as its polymeric assembly within cilia. We define monomer-monomer contacts and membrane-associated regions available for association with transported cargo, and we also use this model to provide insights into the pleiotropic nature of human ciliopathy-associated genetic variants in genes encoding IFT-A subunits. Our work demonstrates the power of integration of experimental and computational strategies both for multi-protein structure determination and for understanding the etiology of human genetic disease.

*For correspondence:
clmccafferty@utexas.edu (CLMcC);
marcotte@utexas.edu (EMM)

Competing interest: The authors declare that no competing interests exist.

## Editor's evaluation

This paper will be of interest to scientists working on cilia, intraflagellar transport, and structural modeling. Using a compelling, integrative modeling approach, the paper provides a fundamental structural model for a part of the molecular machinery that is responsible for cilium assembly.

## Introduction

Cilia are microtubule-based organelles that extend from many eukaryotic cells and play key roles in motility and signaling (*Gigante and Caspary, 2020*). Most motile cilia have a 9+2 microtubule axoneme arrangement (*Luck, 1984*) and are responsible for propelling the cells or adjacent fluid. In contrast, primary cilia lack the central pair of microtubules and serve a key role in cellular sensory functions (*Bloodgood, 2009*; *Garcia-Gonzalo and Reiter, 2012*). In humans, ciliary defects are linked to a number of developmental diseases, broadly known as ciliopathies (*Legendre et al., 2021*; *Reiter and Leroux, 2017*).

Functional cilia require proper assembly of the ciliary axoneme and movement of cargos, both of which are implemented through a process known as intraflagellar transport (IFT). First defined in the flagellated, unicellular green algae *Chlamydomonas* (*Kozminski et al., 1993*), IFT comprises transport of proteins and protein complexes from the cell body to the tip of the cilia by kinesin motors

(anterograde), followed by the movement of molecules back to the cell body by dynein motors (retrograde) (*Lechtreck et al., 2017*; *Taschner and Lorentzen, 2016*). IFT has been shown to transport transmembrane proteins (*Huang et al., 2007*; *Qin et al., 2005*; *Kobayashi et al., 2021*; *Wingfield et al., 2018*), tubulins (*Craft et al., 2015*; *Hao et al., 2011*), and chaperones (*Bhowmick et al., 2009*), among other cargoes necessary for proper ciliary function, although some ciliary proteins also move by diffusion (*Ye et al., 2013*; *Belzile et al., 2013*; *Harris et al., 2016*). Much of our understanding of IFT continues to emerge from work with simple organisms such as *Chlamydomonas*, trypanosomes, and the model single-celled ciliate *Tetrahymena*, but crucially, ciliary assembly and regulation by IFT are strongly conserved across evolution, from unicellular organisms to complex animals (*Ishikawa and Marshall, 2017*).

Accordingly, ciliopathies are a broad class of human disorders arising from dysfunction in cilia and affecting almost all organ systems (*Reiter and Leroux, 2017*). Due to the vital role of IFT in ciliogenesis and maintenance, variants in genes encoding IFT proteins are commonly linked to ciliopathies (*Pigino, 2021*). For example, IFT is critical for primary cilia assembly in mice, where assembly defects are linked to renal disorders, such as polycystic kidney disease (*Pazour et al., 2000*), and IFT-dependent defects in Hedgehog signal transduction produce neural tube and skeletal abnormalities (*Huangfu et al., 2003*; *Haycraft et al., 2007*). Likewise, in clinical studies, variants in human genes encoding IFT components are associated with several skeletal ciliopathies, including Jeune asphyxiating thoracic dystrophy, cranioectodermal dysplasia, and short-rib polydactyly (*Gilissen et al., 2010*; *Ashe et al., 2012*; *Schmidts et al., 2013b*; *Taylor et al., 2015*). Curiously, variants within a single IFT-A gene can be associated with multiple distinct ciliopathies (*Davis et al., 2011*). The nature of such pleiotropy remains unclear.

The IFT complex consists of two distinct subcomplexes, the six-subunit IFT-A complex and the larger, 16-subunit IFT-B (*Taschner and Lorentzen, 2016*). While crystal structures have been determined for several IFT-B subunits (*Taschner et al., 2016*; *Taschner et al., 2014*; *Bhogaraju et al., 2011*; *Wachter et al., 2019*; *Bhogaraju et al., 2013a*; *Taschner et al., 2018*), no such structures exist for any IFT-A proteins. Recently, cryo-electron tomography (cryo-ET) has provided a valuable in situ view of the assembly of IFT complexes into polymeric structures, known as IFT trains, at the ciliary base, highlighting their stepwise association (*Klena et al., 2021*), and capturing snapshots of IFT trains moving along the axonemes (*Jordan et al., 2018*). These cryo-ET studies have not only revealed the low-resolution structures of the IFT-A and IFT-B complexes, but also revealed their cellular context, the mode of IFT-A/IFT-B associations, and the stoichiometries between the complexes. However, this technique has not provided the high-resolution structures necessary to understand the molecular assembly of the individual IFT-A proteins into the overall IFT super-complex.

Here, we present a detailed molecular model of an intact IFT-A complex obtained from a combination of chemical cross-linking mass spectrometry, AlphaFold2 structure and interaction interface prediction, and integrative computational modeling, and by using cryo-ET, we model its assembly into polymeric IFT trains. We validate the model by comparison to experimental evidence across multiple organisms, consistent with the deep structural conservation of the IFT-A complex (*van Dam et al., 2013*). Comparisons to other proteins with related domain architectures show new modes of protein assembly that are unique to IFT-A, as well as preferred interaction modes conserved across protein complexes. Finally, this IFT-A model provides insights into human IFT-A ciliopathy-causing mutations based on their potential to disrupt the IFT-A complex itself or its interaction with cargo. Together, this work highlights the power of integrative modeling in structural biology and provides a mechanistic framework in which to better understand both basic ciliary biology and the complex genotype/phenotype relationships in IFT-associated disease.

## Results

### Determination of individual IFT-A protein structures by AlphaFold2 and chemical cross-linking mass spectrometry

We first independently modeled each of the six individual proteins that constitute the IFT-A complex: IFT43, IFT121, IFT122, IFT139, IFT140, and IFT144 (for simplicity, we use the human gene nomenclature for all genes) (*Behal et al., 2012*). Four of these proteins–IFT121, IFT122, IFT140, and IFT144–are predicted to share the same general architecture of two Tryptophan/Aspartic Acid Repeat (WD40)

head domains and a tetratricopeptide repeat (TPR) tail domain, while IFT139 is composed of 19 TPR repeats, and the IFT43 domain structure is largely uncharacterized (*Behal et al., 2012*). These proteins are broadly conserved across eukaryotes (*van Dam et al., 2013*). While the individual domain structures have been predicted from sequence, with recent advances in structural predictions (*Baek et al., 2021*; *Jumper et al., 2021*), we could model the 3D structures of the full-length proteins with high confidence.

Using the AlphaFold2 Google Colab notebook (*Mirdita et al., 2022*) and protein sequences from the ciliate *Tetrahymena thermophila*, we predicted the structure of each full-length IFT-A protein (*Figure 1—figure supplement 1*), as well as those of *Chlamydomonas reinhardtii* and *Homo sapiens*. The computed models scored well by the predicted local distance difference test (pLDDT) (*Tunyasuvunakool et al., 2021*), with most of the residues falling within the confident prediction threshold (pLDDT >70).

To experimentally validate these structure predictions, we characterized endogenous IFT-A complexes from *Tetrahymena* with cross-linking mass spectrometry (XL/MS), in which cross-linked amino acid residues in a protein sample are connected by covalent crosslinks (XLs) of a defined length and can be identified in MS analyses (*Leitner et al., 2016*; *O'Reilly and Rappsilber, 2018*; *Tang et al., 2021*; *Liu et al., 2017*). Such data define protein interaction interfaces at amino acid resolution and provide distance restraints for structural modeling. Briefly, we purified cilia from *Tetrahymena* (*Rajagopalan et al., 2009*; *Figure 1A*), solubilized the membrane and matrix fraction (M+M) which contains IFT-A and IFT-B (*Lucker et al., 2005*), and then further enriched for monomeric IFT-A complexes using size-exclusion chromatography (SEC) (*Figure 1—figure supplement 2*). We cross-linked the proteins in the IFT-A-containing fractions using a mass spectrometer-cleavable crosslinker, disuccinimidyl sulfoxide (DSSO), and mapped the crosslinked residues using $MS^2/MS^3$ tandem mass spectrometry. DSSO covalently cross-links pairs of accessible lysine residues falling within a distance determined by the length of the DSSO linker itself and the linked lysine side chains (*Merkley et al., 2014*). This is generally less than 30 Å, although we include an additional 5 Å margin of error to account for protein dynamics as in *Erzberger et al., 2014*; *Fernandez-Martinez et al., 2016*; *LoPiccolo et al., 2015*; *Shi et al., 2014*.

We identified a total of 69 intramolecular cross-links between amino acids spanning the length of each individual IFT-A protein, apart from IFT43 (*Figure 1B*, **purple arcs**). We then tested the concordance between these XL/MS data and our AlphaFold2-predicted structures by calculating the distance between each linked residue pair in each predicted IFT-A protein structure. Our XL/MS data strongly validated the predicted structures, as 97% of all cross-linked residue pairs fell within the 35 Å length restraint in the models (*Figure 1C*, **pink in violin plots**). Impressively, even intramolecular cross-linked residues separated by more than 500 residues in their primary sequence fell within 35 Å in the 3D structure (*Figure 1B and C*).

The structures of the IFT-A complex and its subunits are expected to be highly conserved across eukaryotes (*van Dam et al., 2013*), so we explored this relationship by superimposing models of IFT-A proteins from *C. reinhardtii* on those for *T. thermophila*. As expected, our *Tetrahymena*-derived chemical cross-links were still strongly in agreement with the *C. reinhardtii* structures, although as expected the concordance was slightly weaker (*Figure 1C*, **green**). The overall structure was also largely conserved between structure predictions for *Tetrahymena* and human IFT-A proteins (*Figure 1—figure supplement 3*). Accordingly, we observed 86% agreement between the human model and the *Tetrahymena* XL/MS data, measured as the percentage of XL/MS amino acid pairs that were predicted to be 35 Å or less apart when mapped onto the AlphaFold2 models of human IFT proteins. Taken together, the high level of agreement between predicted structures and experimental cross-linking data strongly suggests that the predicted structures accurately capture relevant IFT-A subunit conformations in these highly conserved eukaryotic proteins.

## Integrative modeling of the IFT-A complex

We next sought to build a molecular model of the assembled monomeric IFT-A complex. To this end, we first mined our XL/MS data for intermolecular cross-links, providing distance restraints between pairs of amino acids located in two different IFT-A protein subunits (*Figure 2*, **green lines**). With these new data providing experimentally supported intermolecular contacts, we modeled all possible pairs of IFT-A proteins with AlphaFold-Multimer (*Evans et al., 2021*) to test if the algorithm correctly

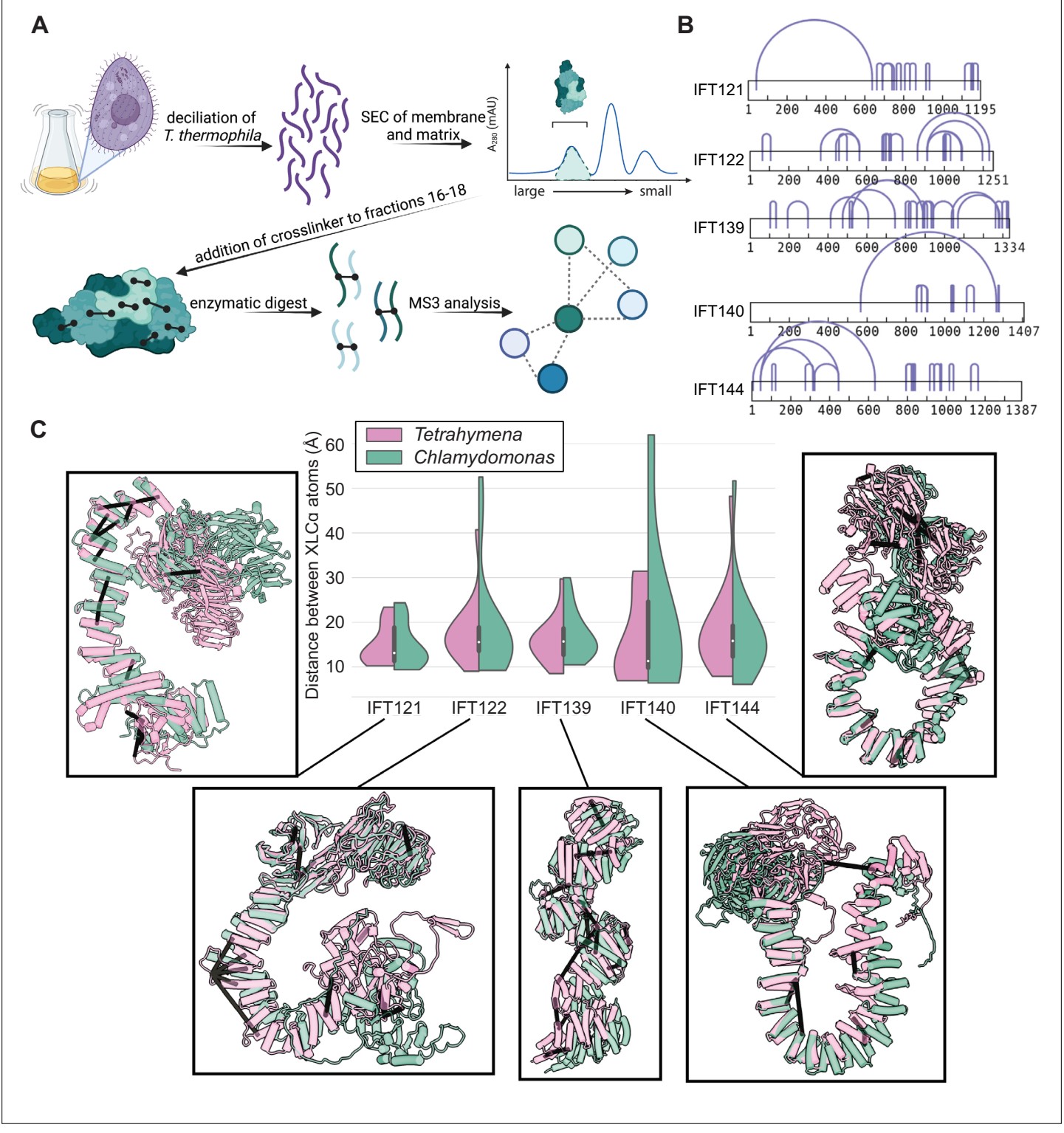

**Figure 1.** AlphaFold2 structures of the IFT-A subunits are supported by experimentally-determined intramolecular cross-links. (**A**) Sample preparation protocol for obtaining an enriched sample of endogenous IFT-A from *T. thermophila*. The sample preparation was followed by chemical cross-linking using DSSO and tandem (MS²/MS³) mass spectrometry to identify cross-linked peptides. Image created with BioRender.com. (**B**) Bar diagrams highlight the extensive intramolecular DSSO cross-links (purple arcs connecting cross-linked amino acid pairs) within each of the IFT-A subunits (bars, numbers denote amino acid positions). (**C**) Violin plots of the distance between Cα atoms of chemically cross-linked residues. Surrounding images show locations of intramolecular crosslinks (black bars) on aligned AlphaFold2 predicted models of IFT-A subunits from *Chlamydomonas reinhardtii* (green) and

*Figure 1 continued on next page*

*Figure 1 continued*

*Tetrahymena thermophila* (pink). A maximum distance of 35 Å between Cα atoms is expected for DSSO cross-links. 97% of intramolecular cross-links are satisfied for *T. thermophila* and 94% for *C. reinhardtii*.

The online version of this article includes the following figure supplement(s) for figure 1:

**Figure supplement 1.** AlphaFold2 structural predictions for *T. thermophila* proteins of the IFT-A complex.

**Figure supplement 2.** Enrichment for monomeric IFT-A from *Tetrahymena* cilia.

**Figure supplement 3.** *Tetrahymena* IFT-A cross-links mapped onto AlphaFold-predicted structures of human IFT-A proteins.

identified the true interaction partners, judged by the crosslinks, and provided 3D models for the interacting domains (*Figure 2—figure supplement 1*). We found near-perfect concordance between AlphaFold-predicted interaction partners and the experimentally confirmed interaction partners, the sole exception being a high-confidence AlphaFold interaction between IFT122 and IFT139 with no supporting cross-links.

Using AlphaFold's predicted aligned error (PAE) confidence scores as a guide (as detailed in *McCafferty et al., 2022b*), we constructed high-confidence models of interacting domains between IFT121-IFT122, IFT121-IFT139, IFT121-IFT43, IFT122-IFT139, IFT122-IFT140, IFT122-IFT144, and IFT140-IFT144 (*Figure 2—figure supplement 2A*). The combination of cross-linked residue pairs, AlphaFold2 monomer structures, and AlphaFold-Multimer domain-domain interaction models provided sufficient spatial restraints to build an initial model of IFT-A, which we improved by using a four-step integrative modeling approach (*Webb et al., 2018*; *Saltzberg et al., 2019*; *Russel et al., 2012*; *Figure 2*, *Figure 2—figure supplement 3*, *Figure 2—figure supplement 4*, and Methods).

We began by gathering data to provide spatial restraints for each of the IFT-A proteins in a manner consistent with prior structural information. We modeled most IFT-A proteins (IFT121, IFT122, IFT139, IFT140, and IFT144) as chains of rigid bodies (*Figure 2—figure supplement 2B*), introducing breaks between individual rigid bodies that corresponded to protein loop regions with lower pLDDT Alpha-Fold2 confidence scores. For IFT43, helices were modeled as rigid bodies, while the remainder of the protein was modeled using flexible beads, as it is thought to be disordered (*Behal et al., 2012*). We then further restrained the model by representing protein interaction interfaces derived from the PAE plots as rigid bodies. We used 98 DSSO chemical cross-links for the modeling, 29 of which were intermolecular, to further refine the model.

Next, we performed an optimized sampling through 20 independent modeling runs, with each run beginning from a unique initial configuration for the model. This exploration of the configuration space (10,000 frames) enabled the IFT-A subunits to find positions that best satisfied the structural restraints. In all, we sampled 200,000 possible configurations for the IFT-A complex. The models were initially clustered based on scores to identify those that best satisfied the input restraints. We selected the cluster of models that best agreed with the chemical cross-linking data. This cluster included models from multiple independent runs that converged on similar arrangements of the IFT-A complex, indicating that the model could be independently determined from different starting configurations (see *Figure 2—figure supplement 3* and **Methods**).

To arrive at a final model, we considered the ensemble of 9,121 models associated with the top-scoring cluster. The cluster had a weighted root-mean-square (RMSF) cluster precision of 15 Å (*Viswanath et al., 2017*), which denotes the average fluctuations of the individual residues (or beads) in 3D space across the ensemble. We visualized these fluctuations across the top-scoring models to define the probability density for each subunit; these probability densities and the centroid of the model ensemble provided our current best estimate of the 3D structure of monomeric IFT-A (*Figure 2*, *Figure 2—figure supplement 3*).

As an initial validation of our IFT-A model, we first examined the agreement with the chemical cross-links. Consistent with the length of the cross-linker and the two coupled lysine side chains, we again considered a cross-link satisfied if the lysine Cα residues were positioned within 35 Å of each other for at least one model in the ensemble. In all, 92% of the cross-links were well-satisfied by our top-scoring ensemble of models (*Figure 2—figure supplement 4*).

We also assessed the quality of the ensemble solution by splitting the models in the final cluster in half and computing the probability densities separately, testing for convergence. We observed a high cross-correlation coefficient between the two samples (.94) confirming a high degree of convergence

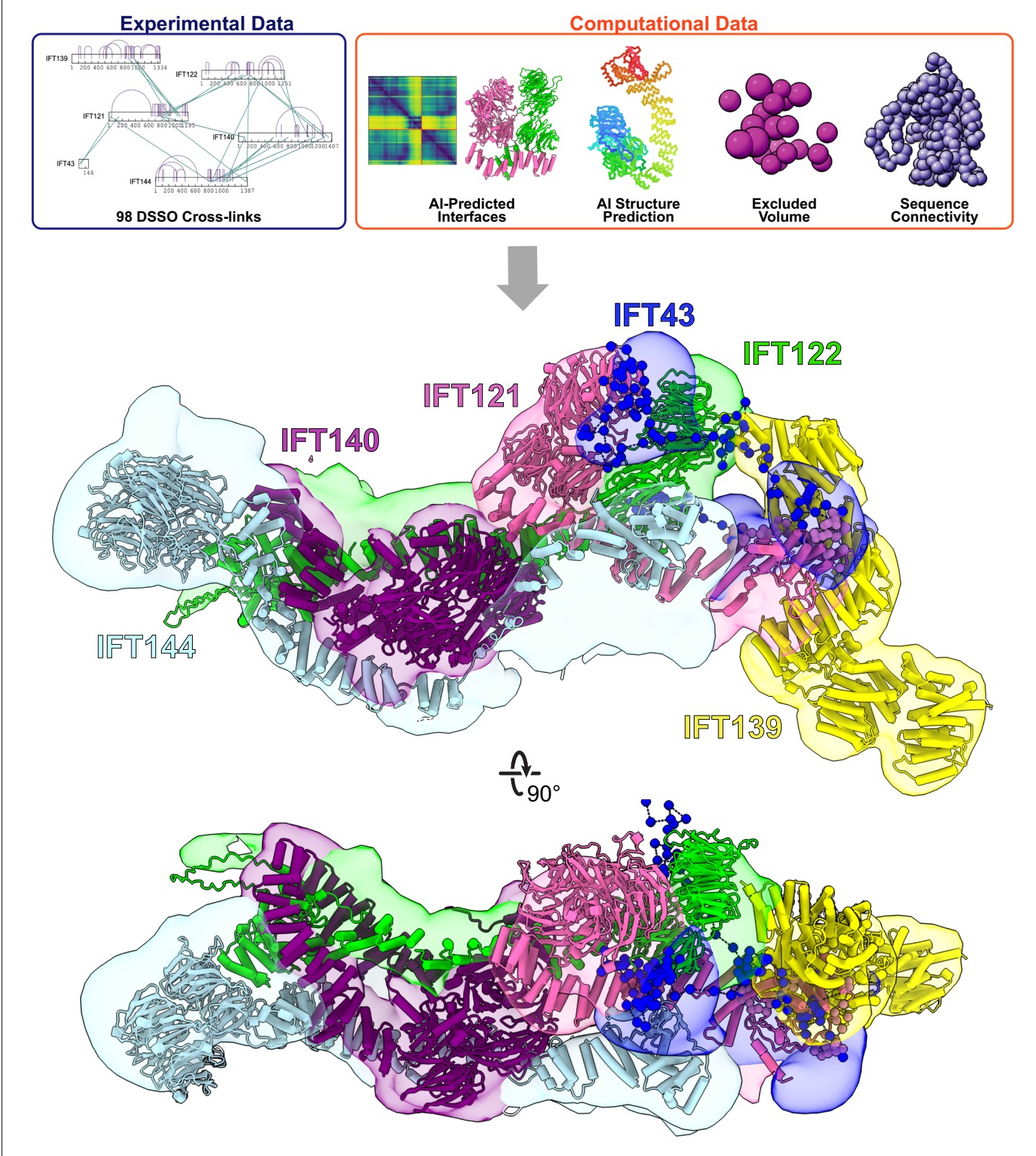

**Figure 2.** Integrative modeling of the IFT-A monomer identifies subunit locations and interactions. Using spatial restraints (**top panels**) based on chemical cross-links (intramolecular, purple arcs; intermolecular, green lines), AlphaFold2 protein models, and AlphaFold2 high-confidence predicted interfaces between proteins (see **Methods** for more details), an integrative model (**bottom panel**) of monomer IFT-A was determined that best satisfied

*Figure 2 continued on next page*

*Figure 2 continued*

these spatial restraints. The centroid model of the top-scoring cluster of 9121 models is shown (ribbon diagram) in the localized probability density (colored volumes) for each subunit.

The online version of this article includes the following figure supplement(s) for figure 2:

**Figure supplement 1.** AlphaFold PAE analysis was used to determine the boundaries for rigid-bodies in integrative modeling.

**Figure supplement 2.** Summary of all AlphaFold-Multimer-predicted intermolecular interfaces used as rigid bodies in the model.

**Figure supplement 3.** Integrative modeling scheme.

**Figure supplement 4.** Initial (**A**) and final (**B**) configurations of the IFT-A monomer model.

between models in our final cluster (*Viswanath et al., 2017*). These values indicate that the selected cluster contains convergent modeling runs in which different starting positions yield highly similar end arrangements of IFT-A subunits. Thus, our structural model of the IFT-A monomer was both strongly internally consistent with the input data from XL/MS and structure predictions and the same solution was repeatedly derived by independent modeling runs.

## Assembly of IFT-A monomers into a polymeric train

As the model was determined primarily using cross-links and structural constraints derived from monomeric IFT-A, we next sought to determine how the monomeric IFT-A might assemble into the polymeric form found in anterograde trains in cilia, and how it might orient with respect to IFT-B and the ciliary membrane. To address these questions, we determined a 23 Å resolution structure of the IFT-A complex by cryo-ET and subtomogram averaging, as observed in situ in the context of a flagellar anterograde IFT transport train within intact *Chlamydomonas* flagella (*Figure 3*). To arrive at this structure, we incorporated 7900 additional particles into the subtomogram averaging from our previous studies (*Jordan et al., 2018*; *Jordan and Pigino, 2019*; *Table 1* and **Methods**), and this improved resolution served to constrain and inform our modeling.

We first performed a rigid body docking of our IFT-A monomeric complex into the subtomogram average using the ChimeraX fit-in-map tool (*Pettersen et al., 2004*). While IFT43, IFT121, IFT139, and the N-terminus of IFT122 fit well into the train, IFT140, IFT144, and the C-terminus of IFT122 required further fitting using molecular dynamics-based flexible fitting (*Kidmose et al., 2019*; *Figure 4*, *Figure 4—figure supplement 1*, *Figure 4—video 1*).

Accommodating the monomeric IFT-A into the anterograde train cryo-ET structure required a rotation of the C-terminus of IFT122 into the neighboring volume, where the extended TPR tail could be fit into a clearly delineated tube of density (*Figure 4—figure supplement 1*). By preserving the relative positions of IFT140/144 relative to the IFT122 C-terminal domain, their positions were also clearly evident in the cryo-ET density, in spite of a large movement of both IFT140 and IFT144 relative to IFT43/121/122 (N-terminus)/139 (*Figure 4—video 1*). Notably, we observed IFT140 rearranged to bridge adjacent IFT-A complexes within the train, with the N-terminus of IFT140 interacting with the C-terminus of an adjacent IFT140 protein. This arrangement was especially interesting in light of AlphaFold's suggestion that IFT140 formed a loop with the N-terminus interacting with its own C-terminus, which may stem in part from evolutionary couplings that reflect the native polymeric state.

## Validation of the modeled IFT-A structure

To further validate our structural model, we investigated whether our model was consistent with the extensive prior biochemical literature identifying direct interactions among the IFT-A proteins. Both co-sedimentation assays and visual immunoprecipitation (VIP) experiments in *Chlamydomonas* and mammalian cells (*Behal et al., 2012*; *Hirano et al., 2017*) suggest that IFT122, IFT140, and IFT144 form a core complex within IFT-A (*Figure 5B*.**I and B.II**). Within this core complex, the C-terminal domain of IFT122 directly interacts with IFT140 and IFT144 (*Takahara et al., 2018*), with a stable heterodimer being formed between IFT122 and IFT144, and specifically, residues 357–653 of IFT144 are required for the interaction with IFT122 (*Hirano et al., 2017*; *Takahara et al., 2018*). Furthermore, visible immunoprecipitation (VIP) assays show the N-terminal domain of IFT122 interacts with IFT121 and IFT43 (*Takahara et al., 2018*). Our model of the IFT-A structure agrees perfectly with these previous findings (*Figure 5B*.**III-5B.VI**).

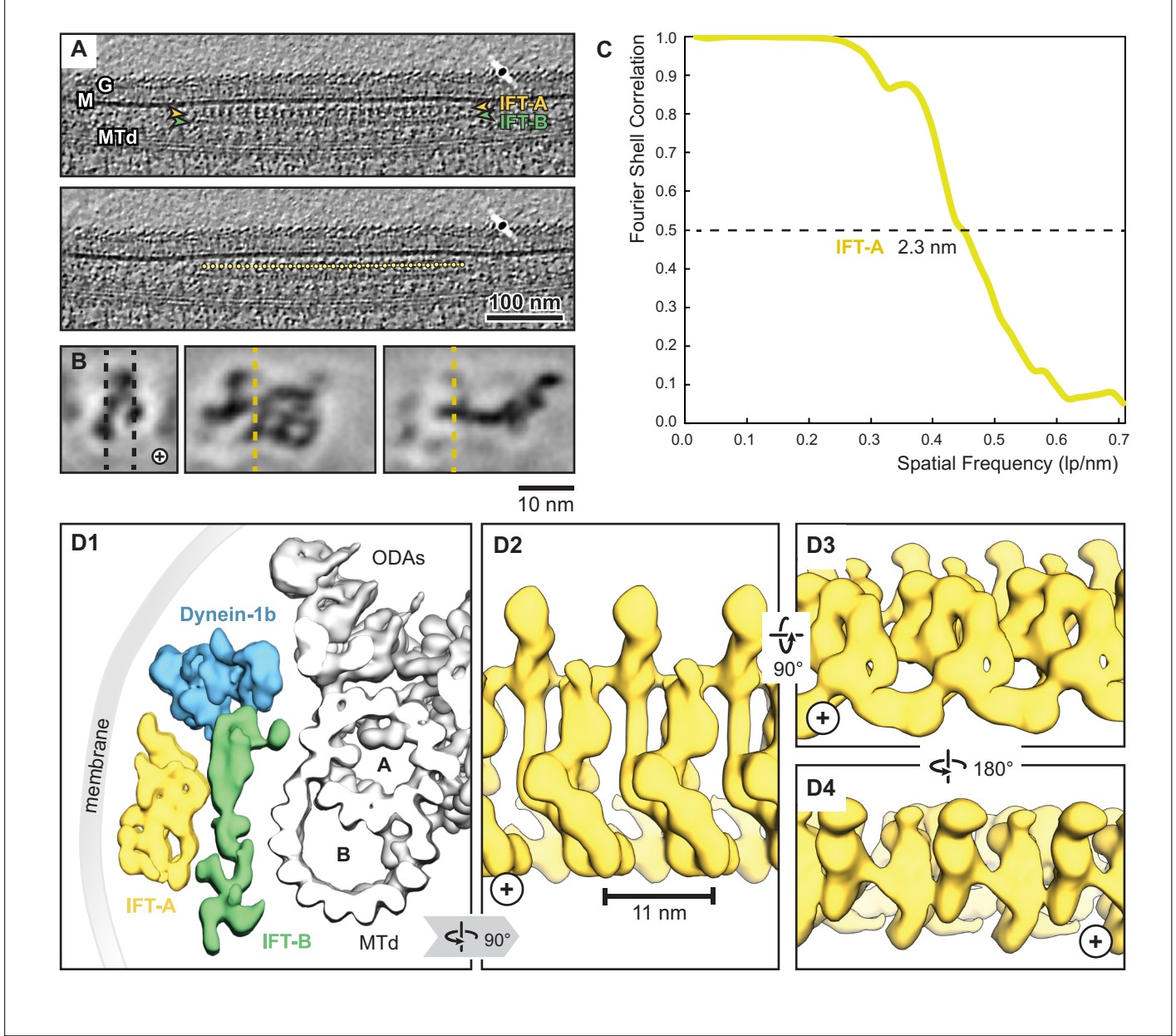

**Figure 3.** Overview of the IFT-A cryo-ET structure. (**A**) IFT train in a raw tomogram of a *C. reinhardtii* cilium with the section showing the repeats of IFT-A and IFT-B. The picking of IFT-A particles is shown in the lower panel (MTd, microtubule doublet; G, glycocalyx; M, membrane; the direction of the ciliary tip is to the right side). (**B**) Subtomogram average of IFT-A; the left panel shows the same orientation as in A (plus, direction of the ciliary tip). Vertical dashed lines in the left panel indicate slice sections corresponding to the middle and right panels, respectively. (**C**) The Fourier shell correlation of the subtomogram average indicates a resolution of 2.3 nm at a cut-off criterion of 0.5. The pixel size is 0.71 nm. (**D1**) 3D isosurface of a reconstructed IFT train in between the membrane and the microtubule doublet as seen from the ciliary base towards the tip (ODAs, outer dynein arms; A, B, A-, and B-tubule) and composed of three averages: IFT-A (yellow), IFT-B (green) and dynein-1b (blue). (**D2**) The IFT-A polymer as seen from the membrane towards the microtubule doublet. (**D3**) and (**D4**) Views of IFT-A as indicated.

Moreover, several studies have cataloged the extensive interactions of IFT121. VIP experiments were used to observe the robust binding of both the IFT121 C-terminal fragment to all other subunits in the complex and the IFT121 fragment 545–800 to IFT122 (*Fu et al., 2016*). Moreover, the interaction between IFT121 and IFT43 was observed for *C. reinhardtii* proteins by yeast two-hybrid assay and recombinant bacterial co-expression (*Behal et al., 2012*). In addition, bacterial co-expression assays demonstrated an interaction between IFT122 and IFT43 (*Behal et al., 2012*). Again, our IFT-A model agreed with all of these independent data (*Figure 5B*.**VII - 5B.X**).

**Table 1.** Cryo-ET data collection and processing statistics.

| | IFT-A complex (EMD-26791) |
|---|---|
| **Data collection** | |
| Magnification | 30,000 |
| Voltage (kV) | 300 |
| Electron exposure (e–/Å²) | 100–140 |
| Defocus range (μm) | –3 to –6 |
| Pixel size (Å) | 14.13 (bin6), 7.08 (bin3) |
| Tilt-range/step (°) | ±64° / 2 |
| **Processing** | |
| Symmetry imposed | C1 |
| Final particle images (no.) | 9350 |
| Map resolution (Å) | 23 |
| FSC threshold | 0.5 |

Finally, several studies focus on direct interactions of IFT139 within IFT-A. One study in mammalian cells using the VIP assay suggests that IFT43, IFT121, and IFT122 interact with IFT139. The study implies that IFT122 may be required for the interaction of IFT139 with IFT43 and IFT121 (*Hirano et al., 2017*). Interestingly, while our data showed no direct cross-links between IFT122 and IFT139, our model suggests that the interaction between the peripheral subunits (IFT43, IFT121, and IFT139) is facilitated by IFT122 (*Figure 5B*.**XI**). It has also been suggested from IFT139 loss-of-function experiments that IFT139 is the most distal subunit (*Hirano et al., 2017*). Consistent with these data, our model places IFT139 most distal to the core (*Figure 5B*.**XII**).

## Discussion

As structural biology moves towards tackling more complicated problems in situ, AI-predicted structures and chemical cross-links provide a complement to cryo-electron microscopy and tomography studies to illuminate the architecture of challenging, transient, or less abundant protein complexes. Here, we have combined cryo-electron subtomogram averaging of intact cilia, chemical cross-links of highly enriched soluble endogenous IFT-A complexes, and AlphaFold2 predicted structures of individual proteins and protein pairs to build a comprehensive 3D model of the IFT-A ciliary trafficking protein complex. The IFT-A structure is strongly supported by previous biochemical interaction studies and reveals new and conserved packing modes among proteins sharing these domain architectures. Moreover, the model provides testable new hypotheses and sheds new light on the precise mechanisms underlying IFT-related human genetic diseases.

### Conserved interactions between structurally similar proteins

We constructed our model using spatial information from organisms that have diverged since the last eukaryotic common ancestor (LECA). Despite this, IFT has proven to be highly conserved (*van Dam et al., 2013*). In one example, there is a 41% sequence identity shared between the IFT172 protein from human and *T. brucei*, an early branching supergroup (Excavata) of eukaryotes (*van Dam et al., 2013*). This structural conservation between species is also evident from IFT-B subtomogram averages collected from mammalian primary cilia and *C. reinhardtii* motile cilia (*Kiesel et al., 2020*), which display similar overall morphologies to the IFT-B monomers.

Furthermore, our new model allows us to compare the protein-protein interactions in IFT-A to those with similar domain architectures to better understand the functions of individual subunits within the complex. Phylogenetic evidence suggests that the IFT complex is a sister structure to COPI and a member of the proto-coatomer family (*van Dam et al., 2013*). The αβ subunits of the COPI complex interact via their TPR tail domain (*van Dam et al., 2013*; *Lee and Goldberg, 2010*), and we observed

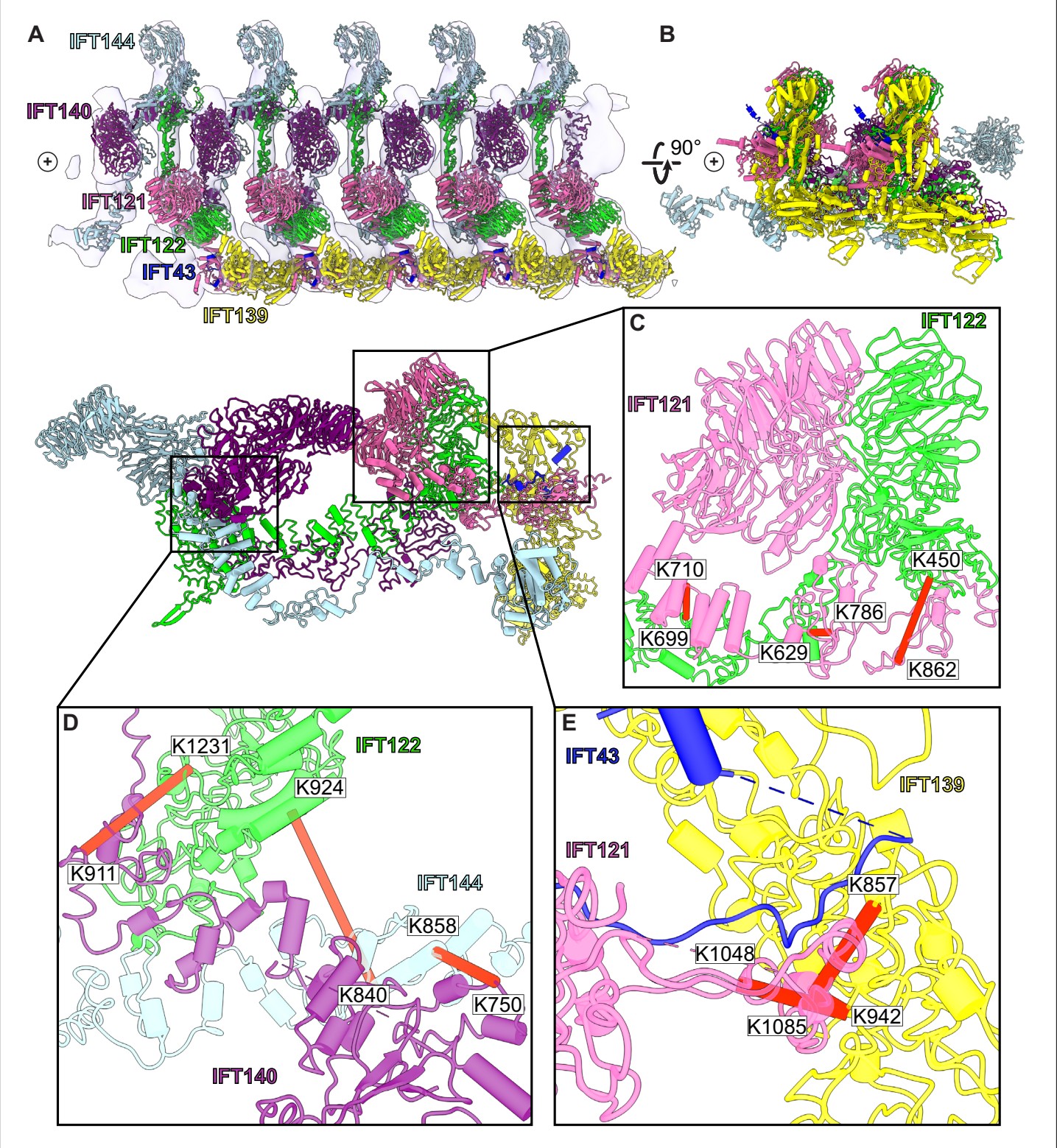

**Figure 4.** A model of the IFT-A train based on docking the monomeric model into the subtomogram average. (**A**) The final cluster centroid model fit into the five IFT-A polymeric repeat subtomogram average map using molecular dynamics-based flexible fitting (*Kidmose et al., 2019*). (**B**) An alternative side view of the IFT-A trains to show interactions between adjacent monomers. Plus signs in A,B indicate the direction of the ciliary tip. (**C**) The IFT121-IFT122 interaction in the train-docked model with satisfied intermolecular cross-linked pairs shown in red. (**D**) The interaction

*Figure 4 continued on next page*

*Figure 4 continued*

between IFT122-IFT140-IFT144 is shown with satisfied intermolecular cross-links in red. (**E**) The interaction between IFT43-IFT121-IFT139 with satisfied intermolecular cross-links between IFT121 and IFT139 shown.

The online version of this article includes the following video and figure supplement(s) for figure 4:

**Figure supplement 1.** Flexible fitting of the IFT-A integrative model into the 23 Å subtomogram average of an anterograde IFT-A train.

**Figure supplement 2.** Independent 3D structural models of IFT-A derived by distinct workflows nonetheless compare favorably.

**Figure 4—video 1.** Fitting monomeric IFT-A into the anterograde train cryo-ET structure.

https://elifesciences.org/articles/81977/figures#fig4video1

a similar binding mode in IFT-A, where the subunits IFT121, IFT122, IFT140, and IFT144 interact *via* their TPR tail domains (*Figure 6*). Phylogenetic analyses also suggest that IFT-A and the BBSome likely evolved from a single IFT complex through subunit duplication (*van Dam et al., 2013*). Interestingly, the BBSome contains β-propeller structures similar to the WD40 domains of IFT-A, in addition to TPR-based subunits similar to IFT139. In the BBSome, the TPR protein, BBS4, interacts with the β-propeller of BBS1 (*Yang et al., 2020*). Similarly, in our model of IFT-A, there is an analogous interaction between the WD40 domain of IFT122 and TPR-based IFT139.

The dominant IFT-A subunit domain structure is two WD40 heads and a TPR tail. Because this domain architecture is so prevalent in the IFT-A complex, we were curious about its representation in the human proteome. We investigated the interactions formed between proteins with the same domain architecture. Sequence-based alignment has traditionally been highly successful in identifying proteins with homologous domains; however, AlphaFold2 has recently enabled new approaches to identify more distantly related homologs that nonetheless retain structural similarity. AlphaFold2 has been used to create databases of proteome-wide structure predictions (*Varadi et al., 2022*), enabling the identification of proteins with similar architectures that are dissimilar in sequence. With this resource now available, an entire proteome can serve as the target of structural similarity searches for discovery of distant homologs, such as by using the program Dali (*Holm and Rosenström, 2010*; *Bayly-Jones and Whisstock, 2021*). We used this approach to compare the four IFT-A proteins sharing the canonical WD40/TPR domain architecture to the set of human proteins with AlphaFold2-predicted structures and found the set of proteins with related structures (*Figure 6—figure supplement 1*). All contained a TPR tail and at least one WD40 head domain.

Analysis of these structurally similar proteins reveals they are generally members of protein complexes involved in transport and trafficking. For example, the elongator complex contains two copies of the ELP1 protein, which interact via their TPR tail domains (*Xu et al., 2015*; *Dauden et al., 2017*; *Figure 6*). Similar to IFT-A, the elongator complex has a role in trafficking (*Rahl et al., 2005*). We identified two of the subunits of the BLOC-2 complex–HPS3 and HPS5–in our Dali search (*Figure 6— figure supplement 1*), as being structurally similar to IFT-A proteins. The BLOC-2 complex is essential for trafficking and is specifically required for the transportation of tyrosinase and Tyrp1 from early endosomes to maturing melanosomes (*Bultema et al., 2012*). Intriguingly, one-third of the mutations leading to Hermansky–Pudlak syndrome occur in BLOC-2 subunits (*Dennis et al., 2015*) suggesting that a better understanding of the complex architecture could impact the treatment of this rare disease. These complexes all have a shared role in cellular transport and trafficking with IFT-A (*Zanetti et al., 2011*; *Popoff et al., 2011*; *Bröcker et al., 2012*).

In addition to overall structural similarity, the interactions among TPR tail domains of these proteins are similar to those of IFT-A (*Figure 6*). This is not surprising given the reported role of TPR domains in stabilizing intracomplex interactions (*Grove et al., 2008*). The recurring interaction between TPR domains raises the question of the role of the associated WD40 domains in these protein complexes. In clathrin, another complex identified in our search, the WD40 domains are known to selectively bind unique cargo peptides in vesicular transport (*ter Haar et al., 2000*). A similar function to this was previously suggested for the WD40 domains in IFT-A and the BBSome (*Bhogaraju et al., 2013b*). We hypothesize that the WD40 domains in these TPR-containing proteins may be specialized for participating in transient interactions such as the binding of cargo proteins and that this role may prove useful for explaining the position of the domains towards the exterior of the complex as well as the range of phenotypes associated with IFT-based diseases.

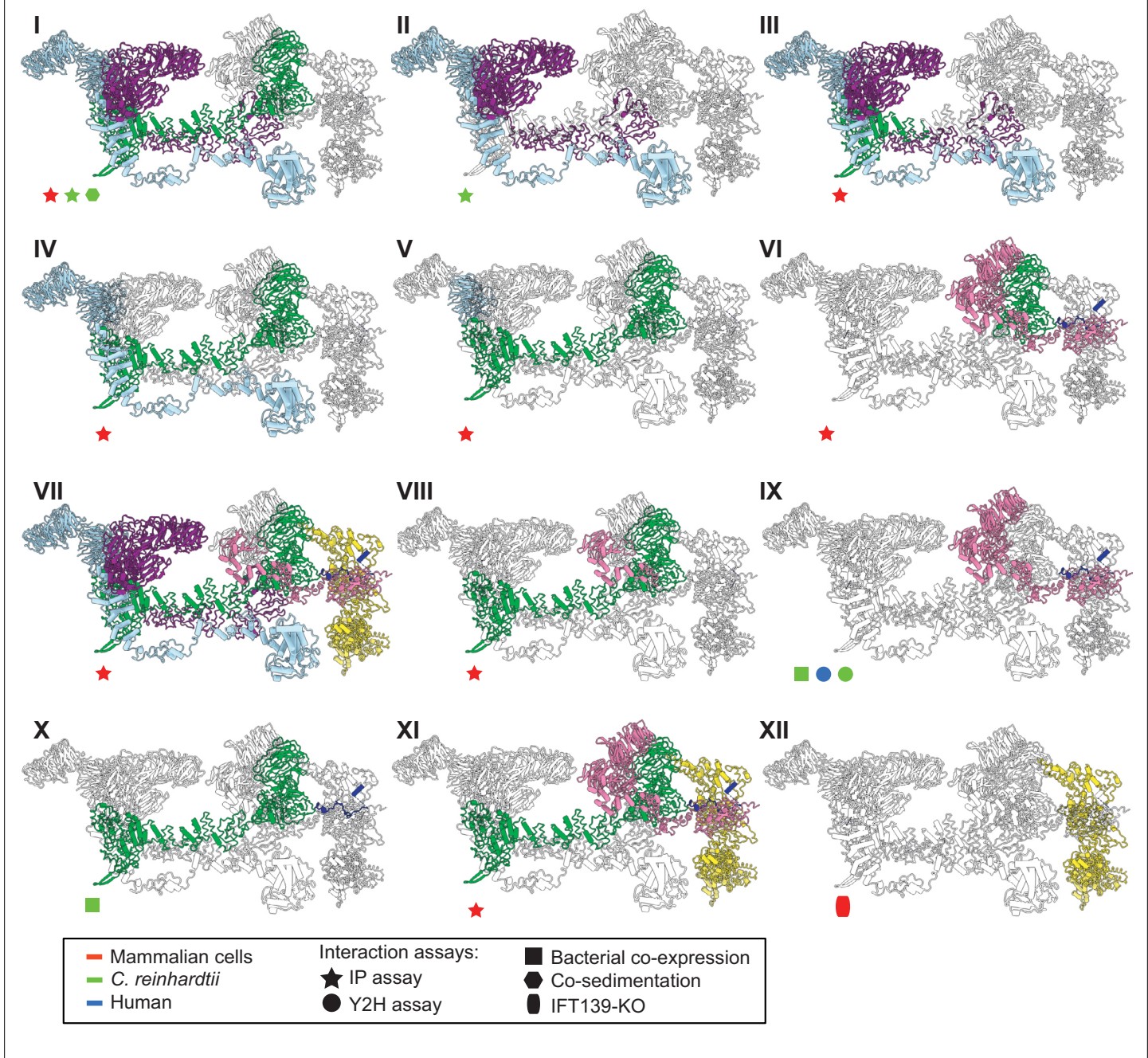

**Figure 5.** The IFT-A 3D model is highly concordant with literature evidence not used in modeling. Panels **I-XII** highlight specific IFT-A intermolecular interactions, both at the protein and domain level, known from the literature (see the text for detailed discussion and citations). The interacting proteins or domains form direct contacts within the IFT-A model in all cases examined. Support for the interactions was observed across organisms, consistent with high conservation of the IFT-A complex across species. IP, immunoprecipitation; Y2H, yeast 2-hybrid; KO, knockout mutant. Proteins are colored as in *Figure 4*.

## Human disease mutations of the IFT-A complex

Variants in the human genes encoding IFT-A subunits are linked to diverse ciliopathies (*Waters and Beales, 2011*), and we examined our IFT-A structure to shed new light on the molecular basis of IFT-A associated diseases. Both the evolutionary history of the IFT-A protein domains (*Grove et al., 2008*) and our new model suggest that the stable formation of the IFT-A complex is mediated by interactions of the TPR domains. While we observe some disease-associated variants occurring within the TPR domains, the large majority of known disease-associated variants (79%) (*Hamosh et al., 2005*;

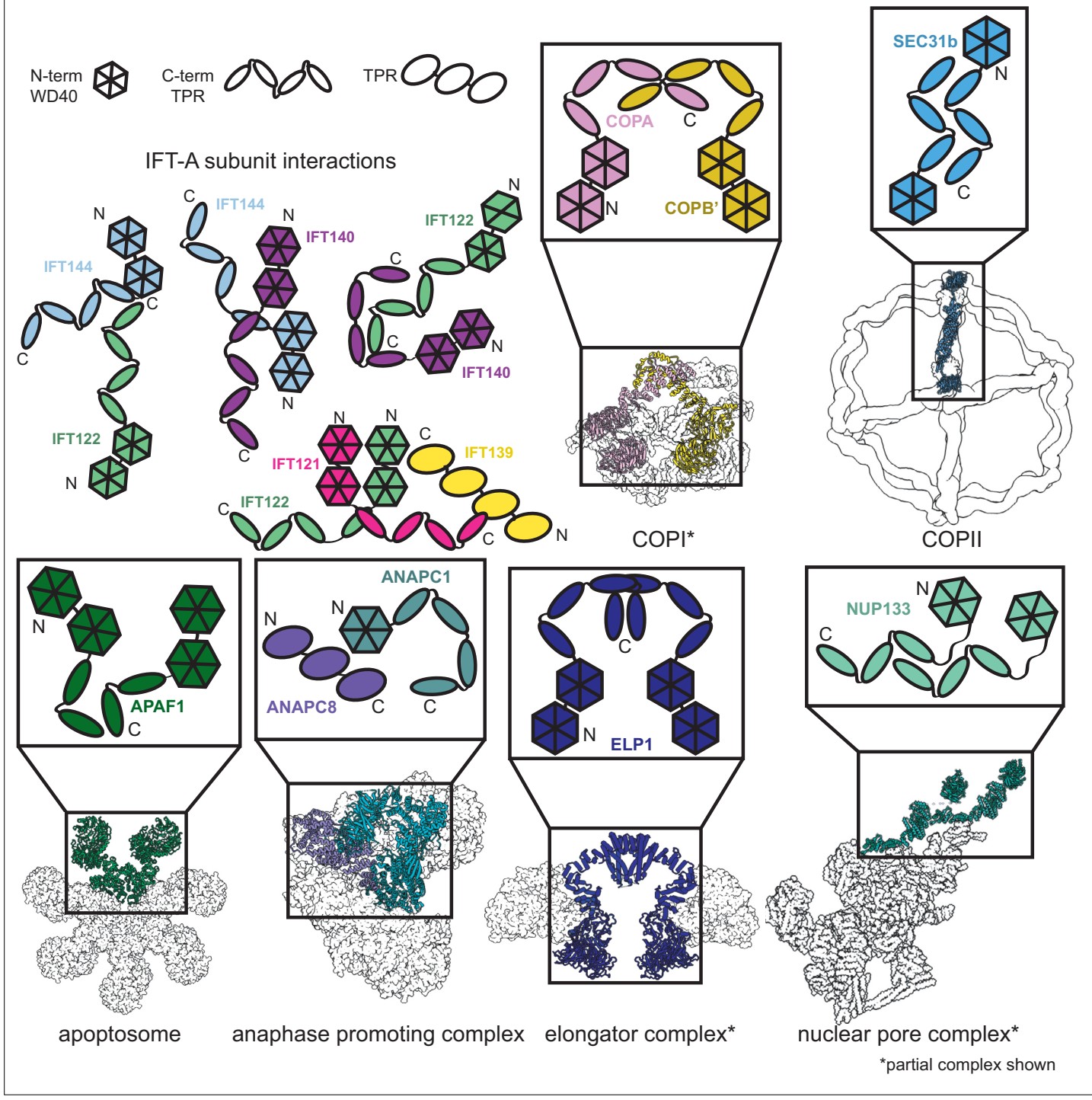

**Figure 6.** Proteins sharing IFT-A subunit domain architectures vary markedly in their quaternary assemblies. By performing structure-based searches (*Holm and Rosenström, 2010*) of the set of AlphaFold2-predicted human protein structures (*Tunyasuvunakool et al., 2021*), we identified proteins that have a similar domain architecture to IFT121, IFT122, IFT140, and IFT144 (as in *Figure 6—figure supplement 1*). Colors highlight the role of these domains in depictions of native complexes containing proteins with these domain architectures. As in our model of the IFT-A complex, stable interactions between TPR domains underlie diverse macromolecular assemblies. COPI PDB: 5A1W (*Dodonova et al., 2015*), COPII PDB: 6ZL0 (*Hutchings et al., 2021*), apoptosome PDB: 5JUY (*Cheng et al., 2016*), anaphase promoting complex PDB: 5G05 (*Zhang et al., 2016*), elongator complex PDB: 5CQR (*Xu et al., 2015*), nuclear pore complex PDB: 7WB4 (*Huang et al., 2022*).

The online version of this article includes the following figure supplement(s) for figure 6:

**Figure supplement 1.** Proteins from the human proteome that are structurally similar to IFT121, 122, 140, and 144.

*Landrum et al., 2014*) are located within the WD40 regions (*Figure 7*). This suggests that the variants located on the WD40 domains act by disrupting the association of important cargo proteins rather than by disrupting overall IFT-A complex formation. This idea is consistent with the proposal that the WD-repeat domains provide large surface areas for potential binding, allowing these domains to simultaneously bind several distinct cargo proteins (*Li and Roberts, 2001*). To explore these ideas, we integrated the known genetics and cell biology of a subset of IFT-A related ciliopathy variants with our new structure of the IFT-A complex (*Figure 8*).

Expression of IFT140 lacking the N-terminal WD repeats is sufficient to disrupt ciliogenesis to some extent but fails to rescue the ciliary localization of GTPases, lipid-anchored proteins, and cell signaling proteins (*Picariello et al., 2019*). This result suggests that the WD40 domains play an important role in cargo transport (*Picariello et al., 2019*) and is consistent with the position of the IFT140 TPR domain in the core of the IFT-A complex and the position of the WD40 domains on the surface of the complex (*Figure 8A*). With this in mind, it is interesting that two disease-associated missense variants in the IFT140 WD40 domains (Y311C and E664K) (*Schmidts et al., 2013a*; *Perrault et al., 2012*; *Figure 8B*) were shown to localize normally to axonemes (*Perrault et al., 2012*), indicating their normal interaction with other IFT-A proteins. By contrast, the S939P variant of IFT40 lies not just in the TPR domain but also near the interaction interface with both IFT122 and IFT144 (*Figure 8D*) and this variant displays an aberrant cellular localization (*Hull et al., 2016*), suggesting failure to assemble into a normal IFT-A complex.

Likewise, loss of IFT122 results in ciliogenesis defects (*Takahara et al., 2018*), but cilia can be rescued by the ciliopathy-associated variants such as the W7C and G513V alleles (*Takahara et al., 2018*), which do not map to the TPR domains but instead to the WD40 repeats (*Figure 8C*). Moreover, cilia rescued with these disease-associated alleles still exhibit abnormal localization of the ciliary proteins INPP5E and GPR161 (*Takahara et al., 2018*), raising the possibility that they specifically disrupt the association of IFT122 with membrane protein cargoes. This is supported by the location of the WD40 domains in our IFT-A structure as sitting under the membrane (*Figure 8A*). These and other ciliopathy-associated missense variants in the WD40 domains of IFT122, including S373F and V553G, also disrupt the association of the IFT-A core with the IFT-A peripheral proteins, IFT121 and IFT43 (*Takahara et al., 2018*), and the positions of all of these alleles near those interfaces in our model are consistent with this result (*Figure 8C*).

The L795P variant in the peripheral component IFT139 is also interesting, as this allele lies in the TPR domain near the protein's interface with IFT121 (*Figure 8E*). The localization of this variant seems normal (*Davis et al., 2011*), yet functional assays in both zebrafish and *C. elegans* demonstrate that the allele is pathogenic suggesting a biochemical perturbation (*Davis et al., 2011*; *Niwa, 2016*). Our model suggests that this allele's pathogenicity may be exerted by disrupting interaction with IFT121 (*Figure 8E*). Thus, our molecular model of the IFT-A complex provides insights into the molecular basis of IFT-A related human ciliopathies.

## Comparison with two independently determined IFT-A structures

Concurrently with our initial preprint (*McCafferty et al., 2022a*), two additional structures were reported for IFT-A particles from other species and determined using different techniques, one for the reconstituted human IFT-A particle determined by single particle cryo-EM (*Hesketh et al., 2022*) and one of the *Chlamydomonas* IFT-A/IFT-B train reconstructed from an 18 Å cryo-ET subtomogram average (*Lacey et al., 2022*). Because our structural constraints–the chemical cross-links and the co-evolutionary couplings incorporated into AlphaFold2–likely represent a combination of IFT-A in the anterograde and retrograde forms, we were interested to compare our model to these contemporaneous structures. While a full comparison will have to await release of atomic coordinates, an initial inspection (*Figure 4—figure supplement 2*) indicates that all three structures are highly concordant in the reasonably compact region of IFT-A defined by the WD40 domains of IFT121/122 and their interactions with IFT139. While IFT43 is absent from *Lacey et al., 2022* and generally highly disordered, our model agrees well with that of Hesketh et al. for the positions of its more ordered segments. Finally, all three IFT-A polymer models appear to agree substantially with regard to the general orientation and placement of the IFT122 extended TPR domain, and to the IFT140-mediated monomer-monomer interactions, placing the N-terminus of IFT140 from one monomer to interact with the C-terminus of an adjacent IFT140 protein. This concordance is especially notable in light

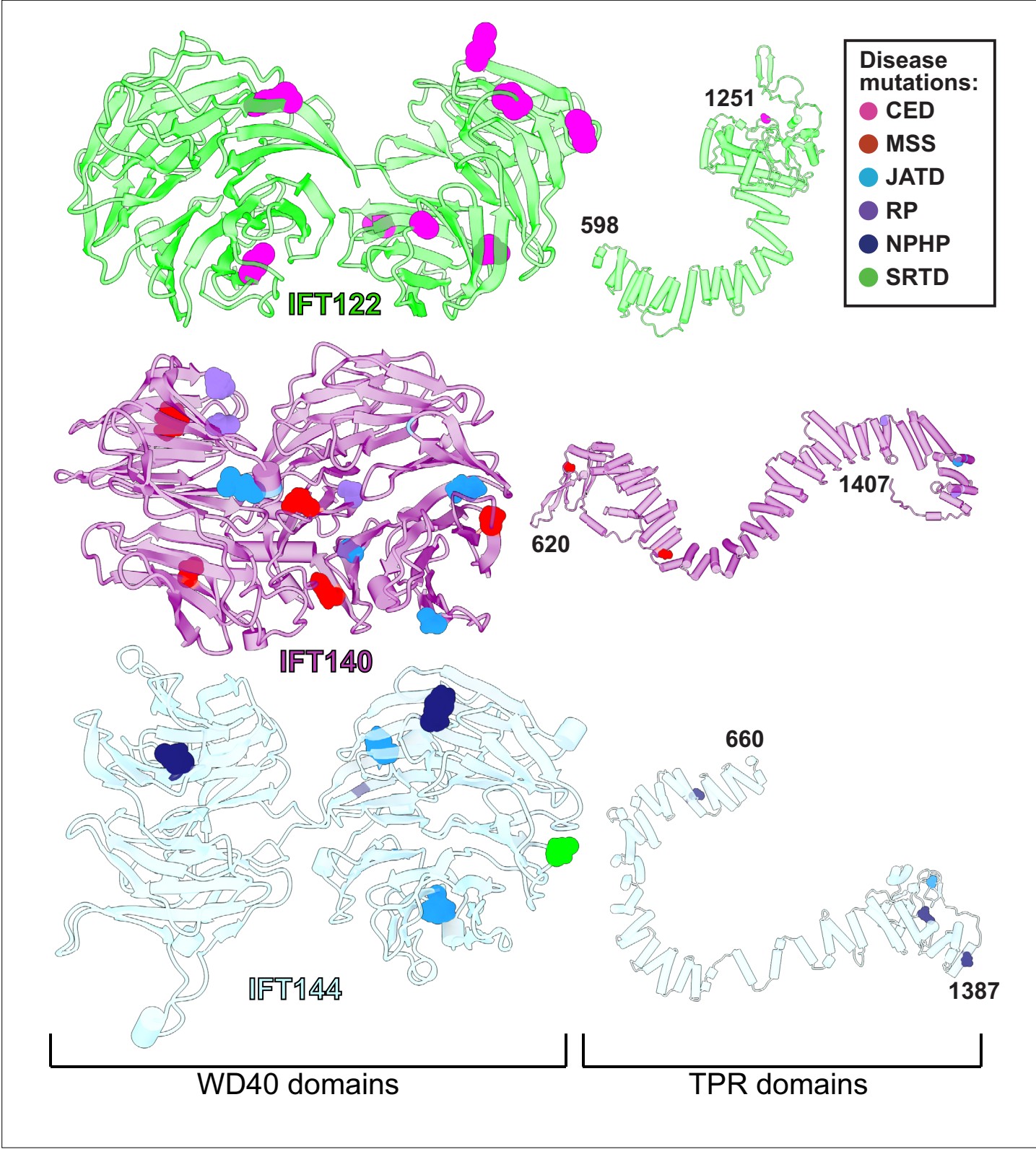

**Figure 7.** WD40 domains of IFT-A core proteins are hotspots for disease-causing missense mutations. Disease-causing missense mutations are displayed on the structures of IFT122, IFT140, and IFT144 (IFT-A core) and colored by disease. 79% of the mutations on the IFT-A core proteins are concentrated in the WD40 domains of these proteins. The mutations are located in the exposed regions of the domains and do not interfere with other IFT-A interactions suggesting they may disrupt more transient interactions formed between IFT-A and its cargos. TPR domains are shown smaller in scale relative to WD40 domains for display purposes. CED, cranioectodermal dysplasia; JATD, Jeune asphyxiating thoracic dystrophy; NPHP, nephronophthisis; MSS, Mainzer-Saldino syndrome; RP, retinitis pigmentosa; SRTD, short-rib thoracic dysplasia.

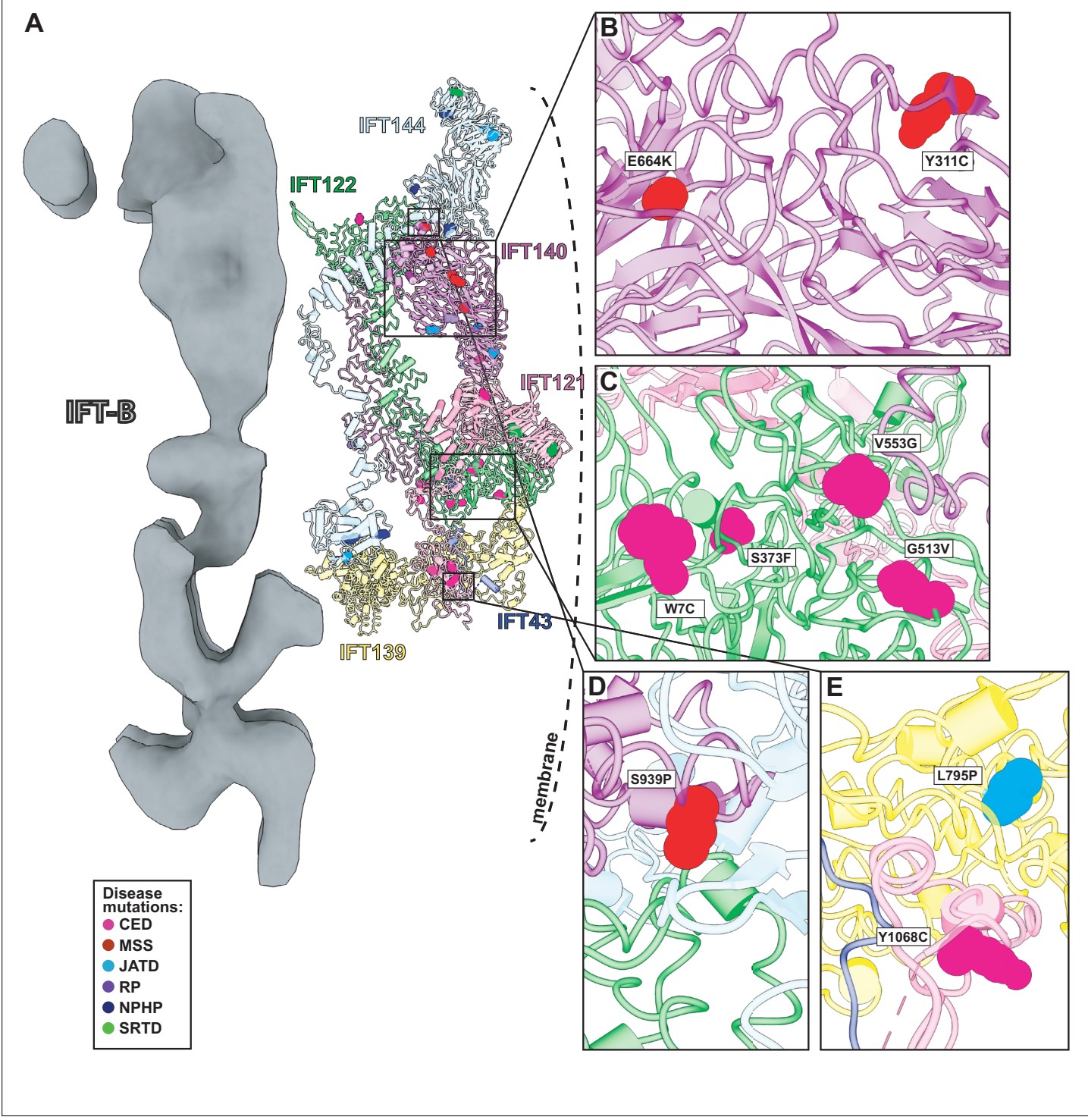

**Figure 8.** Human disease mutations form clusters on exposed WD40 domains and interaction interfaces of the IFT-A complex. (**A**) Human disease mutations are mapped onto our IFT-A structure with missense mutations colored by disease. The dotted line indicates the ciliary membrane. (**B**) The closeup of MSS variants E664K and Y311C, which are located among many other disease-causing variants on the WD40 domains of IFT140. (**C**) A closeup of CED alleles on the WD40 domain of IFT122 are positioned under the membrane and nearby IFT121. (**D**) IFT140 MSS variant S939P lies at the interaction interface with proteins IFT122 and IFT144. (**E**) A closeup of CED variant Y1068C of IFT121 and its proximity to neighboring proteins, IFT43 and IFT139. The closeup also captures JATD variant L795P of IFT139 and its location near the interaction interface with IFT121. Abbreviations are provided in the **Figure 7** legend.

of the different experimental techniques employed by each group and provides some degree of confidence that the structures faithfully capture representative IFT-A conformations. The fact that the combination of cross-linking mass-spectrometry, AlphaFold-Multimer, and integrative modeling produced a highly concordant structure with the significantly higher resolution single particle cryo-EM reconstruction (*Hesketh et al., 2022*) also suggests that such a readily accessible combined modeling approach might be broadly suitable for many other protein complexes.

## Limitations arising from integrative modeling and alternative conformations

A consideration of IFT-A's biological role in ciliary cargo transport is essential in interpreting our model. It has been previously observed that when the IFT trains (including IFT-A, IFT-B, and the BBsome) reach the ciliary tip, a structural rearrangement occurs where kinesin dissociates from the complex and the dynein motors power the retrograde movement (*Chien et al., 2017*). The extent of the rearrangement for the IFT-A complex is not known. However, because our data were derived from IFT complexes solubilized from intact cilia, they should inform models of IFT in both directions. The subtomogram average that we have used in our polymeric model represents the anterograde arrangement and while the retrograde arrangement is not known, it was shown to differ in morphology from the anterograde arrangement (*Stepanek and Pigino, 2016*). It is important to remember that our results represent an ensemble of models that best satisfy the input data, and that some cross-link violations remain for our model (*Figure 2—figure supplement 4*). These remaining violations may point to a possible structural rearrangement in IFT-A proteins during the transition from anterograde to retrograde IFT.

Similarly, the assembly of the IFT-A complex into oligomeric trains clearly suggests structural rearrangements relative to a monomeric IFT-A complex (*Figure 4*, *Figure 4—figure supplement 1*), consistent with observations for the human IFT-A monomer (*Hesketh et al., 2022*). Based on known size calibrants for our SEC column chromatography, our chemical cross-links were derived from the monomeric IFT-A assembly (*Figure 1—figure supplement 2*). However, as the samples were subsequently further concentrated for crosslinking, we speculate that concentrating the particles might have induced some degree of oligomerization and interactions with IFT-B, which may explain the small number of cross-links consistent with IFT-A/A and IFT-A/B interactions. It is unclear, however, if these interactions better reflect particular anterograde or retrograde conformations, and it is possible that additional conformational changes may accompany train assembly or cargo binding that are not captured by these data.

For example, in our monomeric integrative model of the IFT-A complex, we observe a large cross-link violation between the WD40 domain of IFT121 (K16) and IFT144 (K831). Interestingly, while AlphaFold2 places the WD40 domains of IFT121 and IFT122 as interacting and is supported by low PAE values between the residues within this region, we observe no cross-links between the proteins in this region. We do, however, observe cross-links between the WD40 domains of IFT121 and IFT122 and other proteins. These cross-links raise the possibility of domain rearrangements of the WD40 domains of IFT121 and IFT122 upon interactions with other proteins, or even transient rearrangements reflecting a dynamic and flexible complex. As for any structural biology study, while certain regions of the model are highly confident, we anticipate that additional local structural rearrangements are likely to occur with changes to assembly state and direction of travel.

## Methods

**Key resources table**

| Reagent type (species) or resource | Designation | Source or reference | Identifiers | Additional information |
|---|---|---|---|---|
| Gene (*Tetrahymena thermophila*) | IFT43 | Uniprot | Q22NF5 | |
| Gene (*Tetrahymena thermophila*) | IFT121 | Uniprot | Q22U89 | |
| Gene (*Tetrahymena thermophila*) | IFT122 | Uniprot | Q244W3 | |

*Continued on next page*

*Continued*

| Reagent type (species) or resource | Designation | Source or reference | Identifiers | Additional information |
|---|---|---|---|---|
| Gene (*Tetrahymena thermophila*) | IFT139 | Uniprot | I7MFN3 | |
| Gene (*Tetrahymena thermophila*) | IFT140 | Uniprot | I7LVZ7 | |
| Gene (*Tetrahymena thermophila*) | IFT144 | Uniprot | Q22BP2 | |
| Strain, strain background (*Tetrahymena thermophila*) | SB715 | Tetrahymena Stock Center (Cornell University, Ithaca, NY) | SB715 | |
| Software, algorithm | Integrative Modeling Platform (IMP) software | https://github.com/salilab/imp | | |
| Software, algorithm | AlphaFold2 software | Colab version (https://colab.research.google.com/github/sokrypton/ColabFold/blob/main/AlphaFold2.ipynb) | | |
| Software, algorithm | AlphaFold-Multimer | https://github.com/deepmind/alphafold | | |
| Software, algorithm | Namdinator | https://namdinator.au.dk/namdinator/ | | |
| Software, algorithm | SerialEM | https://bio3d.colorado.edu/SerialEM/download.html | | |
| Software, algorithm | K2Align | https://github.com/dtegunov/k2align | | |
| Software, algorithm | IMOD | https://bio3d.colorado.edu/imod/ | | |
| Software, algorithm | ChimeraX | https://www.cgl.ucsf.edu/chimerax/download.html | | |
| Software, algorithm | DigitalMicrograph | https://www.gatan.com/products/tem-analysis/gatan-microscopy-suite-software | | |
| Software, algorithm | Proteome Discoverer 2.3 | Thermo Fisher Scientific | | |

## Tetrahymena culture

*Tetrahymena thermophila* SB715 were obtained from the *Tetrahymena* Stock Center (Cornell University, Ithaca, NY) and grown in Modified Neff medium obtained from the stock center. Cells were routinely maintained at room temperature (~21 ° C) in 10 ml cultures and were expanded to 3 liters at 30 °C with shaking (100 rpm) for preparation of cilia.

## Tetrahymena membrane and matrix preparation

Cilia extracts were made as outlined in *Gaertig et al., 2013*. Briefly, cilia were released by either pH shock or dibucaine treatment and recovered by centrifugation. For cross-linking experiments, cilia were extracted with 1% NP40 in HEPES-Cilia Wash Buffer (H-CWB, where 50 mM HEPES pH 7.4 was used in place of the 50 mM Tris to ensure compatibility with DSSO, and 0.1 mM PMSF was added.) Axonemes were removed by centrifugation at 10,000 x g, 10 min 4 °C. Protein concentration of the soluble membrane and matrix fraction (M+M) was determined by DC BioRad Assay.

## Tetrahymena IFT-A sample preparation

Two types of IFT-A-containing samples were generated: (1) IFT-A enriched fractions from preparative-scale SEC separations, or (2) IFT-A-containing ion exchange chromatography (IEX) fractions. Preparative scale SEC fractionation began with 2.8 mg M+M extract in 2 ml H-CWB. Separation was on a HiLoad 16/600 Superdex 200 PG (preparative grade) column (Cytiva) at a flow rate of 1 ml/min, mobile phase 50 mM HEPES, pH 7.4, 50 mM NaCl, 3 mM $MgSO_4$, 0.1 mM EGTA. 1.5 ml fractions were collected and analyzed by mass spectrometry to confirm the IFT-A elution peak (fractions 16–18, as initially identified from analyses of *Tetrahymena* whole cell extract, corresponding to monomeric IFT-A; see *Figure 1—figure supplement 2*). Molecular mass of eluted IFT-A was estimated using a commercial mixture of molecular weight markers (Sigma-Aldrich #MWGF1000) run under the same conditions (Blue Dextran (approximate molecular mass ~2000 kDa), bovine thyroglobulin, horse spleen apoferritin, bovine serum albumin, and yeast alcohol dehydrogenase) in combination with

eight well-characterized protein complexes of known or inferred molecular mass that were observed in the separations of *Tetrahymena* whole cell extract run under the same conditions. For sample preparation, IFT-A containing fractions from two identical sequential separations were pooled; IFT-A and B were estimated by mass spectrometry to each comprise approximately 3% of the sample. Membrane contamination was removed by centrifugation 100,000 x g 1.25 hours 4 °C in an NVT65.2 rotor (Beckman Coulter). The clarified supernatant was concentrated to 50 µl by ultrafiltration (Sartorius Vivaspin Turbo, 100,000 MWCO) according to manufacturer's instructions. The second set of samples was generated from 1.5 mg of cilia M+M subjected to ultrafiltration (Amicon Ultra Ultracel 10 k NMWL, #UFC501096) to adjust salt and concentrate protein for fractionation using a mixed-bed ion exchange column (PolyLC Inc, #204CTWX0510) with a Dionex UltiMate3000 HPLC system. The chromatographic method was performed as in *McWhite et al., 2021*, but with 10 mM HEPES pH 7.4 replacing Tris in both Buffers A and B.

## Chemical cross-linking / mass spectrometry

Crosslinking was performed on both samples described above: (1) IFT-A enriched SEC fractions and (2) IFT-A-containing IEX fractions. The first sample (representing approx. 40 µg of protein) was cross-linked by addition of DSSO (freshly made 50 mM stock in anhydrous DMSO) to 5 mM final concentration. After 1 hr at room temperature crosslinking was quenched by addition of 1 M Tris pH 8.0–20 mM for 25 min at room temperature. Peptides were reduced, alkylated, digested with trypsin, and desalted using C18 spin tips (Thermo Scientific HyperSep SpinTip P-20 BioBasic # 60109–412) as in *Havugimana et al., 2012* with the exception that reduction was accomplished with 5.0 mM TCEP (Thermo Scientific #77720) instead of DTT. To enrich for cross-linked peptides, the desalted peptides were dried and resuspended in 25 µl 30% Acetonitrile, 0.1% TFA, and separated on a GE Superdex 30 Increase 3.2/300 size exclusion column (Cytiva) at 50 µl/min flow rate using an ÄKTA Pure 25 FPLC chromatography system (Cytiva). A total of 100 µl fractions were collected, dried, and resuspended in 5% acetonitrile, 0.1% formic acid for mass spectrometry.

The second set of IFT-A samples was crosslinked using 50 mM DSSO stock prepared with anhydrous DMF immediately before use, diluted with 10 mM HEPES pH 7.4, and added to each 0.5 ml column fraction to a final concentration of 0.5 mM. Cross-linking proceeded for 1 hr at room temperature and was quenched by addition of 1 M Tris pH 8.0–28 mM. Samples were prepared for mass spectrometry using Method 1 from *McWhite et al., 2021* with no enrichment for crosslinked peptides.

Mass spectra were collected on a Thermo Orbitrap Fusion Lumos tribrid mass spectrometer. Peptides were separated using reverse phase chromatography on a Dionex Ultimate 3000 RSLCnano UHPLC system (Thermo Scientific) with a C18 trap to Acclaim C18 PepMap RSLC column (Dionex; Thermo Scientific) configuration. Data were collected from an aliquot of the cross-linked peptides prior to SEC enrichment using a standard top speed HCD MS1-MS2 method (*McWhite et al., 2020*) and analyzed using the Proteome Discoverer basic workflow, and the proteins identified served as the reference proteome for subsequent cross-link identification from the cross-link-enriched fractions.

To identify DSSO cross-links, spectra were collected as follows: peptides were resolved using a 115 min 3–42% acetonitrile gradient in 0.1% formic acid. The top speed method collected full precursor ion scans (MS1) in the Orbitrap at 120,000 m/z resolution for peptides of charge 4–8 and with dynamic exclusion of 60 s after selecting once, and a cycle time of 5 sec. CID dissociation (25% energy 10 msec) of the cross-linker was followed by MS2 scans collected in the orbitrap at 30,000 m/z resolution for charge states 2–6 using an isolation window of 1.6. Peptide pairs with a targeted mass difference of 31.9721 were selected for HCD (30% energy) and collection of rapid scan rate centroid MS3 spectra in the ion trap. Data were analyzed using the XlinkX node of Proteome Discoverer 2.3 and the XlinkX_Cleavable processing and consensus workflows (*Liu et al., 2017*) and results exported to xiView (*Graham et al., 2019*) for visualization.

## Chlamydomonas cell culture

*Chlamydomonas* wild-type cells (CC-124 mt- and CC-125 mt+) were obtained from the *Chlamydomonas* resource center (https://www.chlamycollection.org); ift46-1::IFT46-YFP was a gift of K. Huang and G. Witman (*Lv et al., 2017*). Cells were cultured in TAP (Tris-Acetate-Phosphate) medium as described by the resource center. For long-term storage, cells were grown on TAP plates with 1.5%

agar at room-temperature. For sample preparation, fresh liquid cultures of 300 mL were grown for three to four days at 22 °C under a light-dark cycle with constant aeration.

## Preparation of cryo-TEM grids by plunge freezing

TEM gold grids with Holey Carbon support film (Quantifoil Micro Tools GmbH, Au 200 mesh R3.5/1) were glow-discharged in a plasma cleaner (Diener electronic GmbH, Femto). For plunge freezing, a Leica EM Grid Plunger (GP) was used at 18 °C and a humidity of ~80%. 3 µL undiluted *Chlamydomonas* cells were applied to the grid and mixed with 1 µL 10 nm colloidal gold particles (BBI solutions). Blotting was performed from the back for 1 s and the grid was plunged into liquid ethane at –182 °C and stored in liquid nitrogen until data acquisition.

## Cryo-ET data acquisition

For data acquisition, a Thermo Fisher (former FEI) Titan Halo cryo-TEM was used, equipped with a field emission gun (FEG), operating at 300 kV. Images were recorded on a K2 summit direct electron detector (Gatan) with an energy filter (GIF, Gatan image filter). Digital Micrograph software (Gatan) was used to tune the GIF, and SerialEM software (*Mastronarde, 2005*) was employed for the automated acquisition of tomographic tilt series in low-dose mode. Tilt series were acquired at a magnification of 30,000 X with a pixel size of 0.236 nm in super-resolution mode of the K2 camera. The tilting scheme was bidirectional with a starting angle of –20° and maximal tilt angles of ±64° when possible. Images were acquired every 2 degrees. The defocus range was between –3 and –6 µm, the slit width of the energy filter was 20 eV. Each tomogram had a cumulative dose between 100 and 140 $e^-/Å^2$ with an image dose of 1.8–2.1 $e^-/Å^2$. Exposure times were between 1.6 and 2.5 s per image, while each 10 frames were acquired. The sample drift was held well below 1 nm/s, and 34 grids were imaged in all. Data collection and refinement statistics are summarized in *Table 1*.

## Tomographic reconstruction

Frame alignment was done with K2Align (provided by the Baumeister group, MPI for Biochemistry, Munich), which is based on the MotionCorr algorithm (*Li et al., 2013*). Tomograms were reconstructed with Etomo from IMOD (*Kremer et al., 1996*), using fiducial markers for alignment. CTF curves were estimated with CTFPLOTTER and the data were corrected by phase-flipping with CTFPHASEFLIP, both implemented in IMOD (*Xiong et al., 2009*). Dose weight filtration was applied (*Grant and Grigorieff, 2015*) and tomograms were reconstructed by weighted back-projection and subsequently binned by 3 and 6, resulting in pixel sizes of 0.708 nm and 1.413 nm.

## Subtomogram averaging

IFT-A particles were picked and averaged as described in detail previously (*Jordan and Pigino, 2019*). The data from wild-type cells (CC-124 and CC-125) was complemented with data from IFT46-YFP cells, which showed no differences in their IFT-A structure compared to wild-type. In short, subtomogram averaging was performed with PEET version 1.11.0 from the IMOD package (*Heumann et al., 2011*). Particles were picked with 11 nm spacing and pre-aligned to a reference generated from particles of one train on bin6 tomograms. The alignment was then refined on bin3 tomograms. To reduce the influence of IFT-B or the membrane, loose binary masks were applied to the reference. The final average was calculated from 9,350 particles derived from 96 tomograms. The resolution was estimated to 2.3 nm by Fourier shell correlation, using a cut-off criterion of 0.5. A reconstruction of the whole IFT-A polymer was generated in UCSF Chimera by placing several copies of one IFT-A unit along the train polymer with 'Fit in map' and merged with 'vop maximum'.

## Modeling of IFT-A protein subunits and pairwise interactions

The IFT-A complex is composed of IFT43, IFT121, IFT122, IFT139, IFT140, and IFT144. Each subunit was modeled independently using the ColabFold notebook (*Mirdita et al., 2022*) with AlphaFold2 (*Jumper et al., 2021*). Modeling results and statistics show that confident structures were generated for five of the six protein subunits (*Figure 1—figure supplement 1*). Pairwise interacting protein structures were predicted using the 2.1.2 version/release of AlphaFold-Multimer (*Evans et al., 2021*) as implemented on Texas Advanced Computing Center (TACC) Maverick2 and Frontera (*Stanzione et al., 2020*) GPU computer clusters. Predicted aligned error (PAE) plots were used as in *McCafferty*

*et al., 2022b* to determine interaction interfaces to be represented as rigid bodies in integrative modeling. We use the calibration curve from our previous studies to select pairwise interactions with PAE of less than 3.5 Å.

## Domain representation and spatial restraint configuration

We used the Python modeling interface of the Integrative Modeling Platform (*Webb et al., 2018*) to model the IFT-A complex, performing the modeling in four stages: (1) gathering data, (2) domain and spatial restraint representation, (3) system restraint and restraint scoring, and (4) model validation (*Saltzberg et al., 2019*; *Russel et al., 2012*; *Ganesan et al., 2020*). Because there are no available crystal structures for any of the IFT-A subunits, we used AlphaFold2 structural models to construct representations of each of the subunits. IFT139 has a long alpha solenoid domain structure, which was independently supported by 18 cross-links, and it was represented as two rigid bodies. In contrast, IFT43, with its more poorly defined structure, was represented as two alpha helices connected by a flexible string of 1 Å beads. IFT121, IFT122, IFT140, and IFT144 share a similar domain architecture of two WDR domains and an alpha solenoid tail domain. All four of these proteins were represented as chains of rigid bodies that corresponded to regions of high AlphaFold2 confidence scores (high pLDDT scores). For pairs of IFT-A proteins, we used PAE estimated errors (*Figure 2—figure supplement 1*) to determine boundaries of rigid body interactions between pairs of IFT-A proteins predicted by AlphaFold-Multimer to interact. The initial model was built from combining the pairwise AlphaFold-Multimer structure predictions into a consensus model, thus preserving the arrangements of predicted interfaces (*Figure 2—figure supplement 2* and *Figure 2—figure supplement 4*).

In total, we considered 98 inter- and intramolecular cross-links, representing the combined cross-linking evidence from two alternative IFT-A enrichment procedures. Cross-links were modeled in IMP using a length of 21 Å as in *Erzberger et al., 2014*. An excluded volume restraint was incorporated to 10 residue beads ensuring that two volumes do not occupy the same space, and a connectivity restraint was applied between beads to ensure that consecutive protein segments remained nearby in 3D space. The full set of restraints was used in creating the scoring framework for the model, and all relevant data and scripts are available on the Zenodo repository.

## System sampling, scoring of restraints, and initial model validation

The 10 rigid bodies were randomized into their initial configurations. A steep gradient descent minimization based on connectivity was used to ensure that neighboring residues were close to each other before Monte Carlo sampling. We then performed 20 independent runs of Monte Carlo sampling, each run starting from a unique initialization configuration and sampling 10,000 frames, thus sampling 200,000 total configurations. Ensembles of models were then clustered first based on cross-linking agreement, sequence connectivity, enclosed volume, and total score. We selected the cluster with models from multiple runs showing high agreement with the cross-linking data and high overall scores for subsequent analyses. The top cluster was assessed against the input data and tested for convergence using the sampling exhaustiveness protocol (*Viswanath et al., 2017*).

## Docking and polymer modeling

Our integrative IFT-A monomer model was rigidly docked into the IFT-A train using the ChimeraX fit in map tool (*Goddard et al., 2018*). Rigidly docking the model placed 54% of the atoms within the map. IFT43, IFT121, IFT139, and the C-terminus of IFT122 subcomplex fit better into the subtomogram average with 67% of the atoms within the map. The remainder of the IFT-A integrative model was fit into the map by successive rounds of breaking the structures and refining their fit using Namdinator (*Kidmose et al., 2019*). The final flexibly fit structure places 71% of all atoms within the map.

## Data deposition

Mass spectrometry proteomics data was deposited in the MassIVE/ProteomeXchange database (*Deutsch et al., 2020*) under accession number PXD032818. Cryo-tomography data was deposited in the Electron Microscopy Data Bank (*Patwardhan, 2017*) under accession number EMD-26791. IFT-A models were deposited in the PDB-Dev database (*Burley et al., 2017*) as well as on Zenodo at doi: 10.5281/zenodo.7222413, along with additional supporting materials, including integrative modeling data and code.

## Acknowledgements

The authors gratefully acknowledge suggestions from Anthony Roberts (Birkbeck/UCL) and the two anonymous referees that substantially improved the paper, as well as Jaime Hibbard for helpful discussion, Agnes Toth-Petroczy (MPI-CBG, Dresden) for early discussions about homology modeling, and the generous support of the *Tetrahymena* stock center (Cornell University) and *Chlamydomonas* resource center that made this project possible. Research was funded by grants from the National Institute of General Medical Sciences R35GM122480 (to EMM) and R35GM138348 (to DWT), National Science Foundation (2019238253 to CLM), National Institute of Child Health and Human Development (HD085901 to JBW and EMM), Army Research Office (W911NF-12-1-0390 to EMM), Welch Foundation (F-1515 to EMM, F-1938 to DWT), and Max Planck Society (MAJ and GP). DWT is a CPRIT Scholar supported by Cancer Prevention and Research Institute of Texas (RR160088). The authors would like to thank the Electron Microscopy Facility of the MPI-CBG. The authors acknowledge the Texas Advanced Computing Center at The University of Texas at Austin for providing high-performance computing resources that have contributed to the research results reported within this paper.

## Additional information

### Funding

| Funder | Grant reference number | Author |
| --- | --- | --- |
| National Science Foundation | 2019238253 | Caitlyn L McCafferty |
| National Institute of General Medical Sciences | R35GM122480 | Edward M Marcotte |
| National Institute of General Medical Sciences | R35GM138348 | David W Taylor |
| National Institute of Child Health and Human Development | HD085901 | John B Wallingford Edward M Marcotte |
| Army Research Office | W911NF-12-1-0390 | Edward M Marcotte |
| Welch Foundation | F-1515 | Edward M Marcotte |
| Welch Foundation | F-1938 | David W Taylor |
| Max Planck Society | | Mareike A Jordan Gaia Pigino |
| Cancer Prevention and Research Institute of Texas | RR160088 | David W Taylor |
| Human Technopole | | Gaia Pigino |

The funders had no role in study design, data collection and interpretation, or the decision to submit the work for publication.

### Author contributions

Caitlyn L McCafferty, Conceptualization, Formal analysis, Investigation, Methodology, Writing – original draft, Writing – review and editing; Ophelia Papoulas, Formal analysis, Investigation, Methodology, Writing – original draft, Writing – review and editing; Mareike A Jordan, Formal analysis, Investigation, Methodology, Writing – review and editing; Gabriel Hoogerbrugge, Candice Nichols, Conceptualization; Gaia Pigino, Writing – original draft, Resources, Writing – review and editing; David W Taylor, Resources, Methodology; John B Wallingford, Investigation, Methodology; Edward M Marcotte, Writing – original draft, Resources, Investigation, Writing – review and editing, Formal analysis

### Author ORCIDs

Caitlyn L McCafferty http://orcid.org/0000-0002-0872-4527
Ophelia Papoulas http://orcid.org/0000-0002-6370-0616
Mareike A Jordan http://orcid.org/0000-0001-6248-8863

Gabriel Hoogerbrugge http://orcid.org/0000-0001-7617-3168
Gaia Pigino http://orcid.org/0000-0002-2295-9568
David W Taylor http://orcid.org/0000-0002-6198-1194
John B Wallingford http://orcid.org/0000-0002-6280-8625
Edward M Marcotte http://orcid.org/0000-0001-8808-180X

## Decision letter and Author response

Decision letter https://doi.org/10.7554/eLife.81977.sa1
Author response https://doi.org/10.7554/eLife.81977.sa2

# Additional files

## Supplementary files

• MDAR checklist

## Data availability

Mass spectrometry proteomics data was deposited in the MassIVE/ProteomeXchange database (*Deutsch et al., 2020*) under accession number PXD032818. Cryo-tomography data was deposited in the Electron Microscopy Data Bank (*Patwardhan, 2017*) under accession number EMD-26791. IFT-A models were deposited in the PDB-Dev database (*Burley et al., 2017*) as well as on Zenodo at DOI: https://doi.org/10.5281/zenodo.7222413, along with additional supporting materials, including integrative modeling data and code.

The following datasets were generated:

| Author(s) | Year | Dataset title | Dataset URL | Database and Identifier |
| --- | --- | --- | --- | --- |
| Marcotte E | 2022 | DSSO Crosslinking of Partially Purified IFT-A from Tetrahymena thermophila | http://proteomecentral.proteomexchange.org/cgi/GetDataset?ID=PXD032818 | ProteomeXchange, PXD032818 |
| Marcotte E | 2022 | Cryo-EM subtomogram average of IFT-A in anterograde IFT trains at 23 Angstrom resolution (Chlamydomonas reinhardtii) | https://www.ebi.ac.uk/emdb/EMD-26791 | Electron Microscopy Data Bank, 26791 |
| Marcotte E | 2022 | An integrative modeling approach reveals the 3D structure of the Intraflagellar Transport A (IFT-A) complex | https://zenodo.org/record/7222413#.Y3NWy-zMLyg | Zenodo, 10.5281/zenodo.7222413 |

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
