## [Editor Report]

This paper will be of interest to scientists working on cilia, intraflagellar transport, and structural modeling. Using a compelling, integrative modeling approach, the paper provides a fundamental structural model for a part of the molecular machinery that is responsible for cilium assembly.

---

## [Decision Letter]

**Decision letter after peer review:**

Thank you for submitting your article "Integrative modeling reveals the molecular architecture of the Intraflagellar Transport A (IFT-A) complex" for consideration by *eLife*. Your article has been reviewed by 2 peer reviewers, and the evaluation has been overseen by a Reviewing Editor and Suzanne Pfeffer as the Senior Editor. The reviewers have opted to remain anonymous.

Essential revisions:

The reviewers are concerned that there are very large differences between this model and the much higher resolution experimentally derived structures. To help you to re-evaluate your methodology and conclusions they suggest:

1. Need documentation of the preparation used (SEC profile, SDS-PAGE, etc.) before and after X-linking. One reviewer thinks it is likely that IFT-A will no longer form a chain, so they are less concerned about A-to-A interactions in the XL experiments.

2. Need to perform integrative modeling using sub-complexes predicted by α fold to get a better-corrected model. The fact that the model in part deviates from data presented in other preprint studies is concerning.

3. Need to address deeply the similarities/differences of the cryo-ET map of Lacey et al. and other IFT-A models.

*Reviewer #1 (Recommendations for the authors):*

1) From personal experience I know how difficult it can be to fit multiple subunits into low-resolution maps (manually or automatically) as completely different solutions often provide equally good fits to the map. This exercise becomes increasingly easier the fewer and larger the sub-complexes are. In the current study, the AlphaFold multimer would be an excellent resource to model dimeric and trimeric IFT-A sub-complexes, which could then be used as a starting point for the structural modeling of the complete IFT-A complex.

*Reviewer #2 (Recommendations for the authors):*

I wonder if the Report format is optimal for this paper or if the traditional separation of results and discussion would be more appropriate. I also think the manuscript could be strengthened by including some documentation (silver stained gel or general MS) showing the IFT-A raw material used for the cross-linking experiments (or a similar prep), which is described only as "highly enriched endogenous IFT-A complexes". While the data show that the approach worked it would nevertheless be informative to know the relative enrichment of IFTA, which will inform on the robustness of the method and could be also helpful to apply the approach developed here to other particles. Also, explanatory figure legends need to be provided for the supplementary figures.

"Accordingly, we observed 86% agreement between the human model and the Tetrahymena XL/MS data."

Explain "86% agreement". Does it mean that the X-linked residues in Tetrahymena are predicted to be 35 nm or less apart in human IFT proteins by AlphaFold2?

"9 + 2 microtubule axoneme, a microtubule-based cytoskeletal structure".

"followed by the movement of products back to the cell body by dynein motors"

Not sure what is actually produced in cilia, products are potentially misleading.

"although some cargoes also move by diffusion (17,18)."

Some "ciliary proteins" might be a better term as "cargo" implies transport by a motor-based process. Add Harris et al. MBoC to references for diffusion.

"inter-molecular" and "intermolecular" are both used.

"We then selected the cluster that best agreed with the chemical cross-linking data with multiple runs converging on similar arrangements of the IFT-A complex, indicating that the model arrived independently from different starting configurations (see Figure S3, and Methods)."

The sentence is not clear to me. I wonder if it would make sense to show some more deviant models/clusters with scores (how many x-links still fit within 3.5 nm?)

Legend Figure 1B: Bar diagrams.

Figure 4, legend: "For comparison, two models, each derived from half the data, exhibited a cross-correlation coefficient of 0.90."

Meaning what? If the particle is cut in half, each half fits better into the outline based on cryo-EM.

Legend Figure 5: I think the legend should be extended explaining the color code for the different species. The Roman numbered subpanels are explained in the text but a shorter version should be added to the legend.

P4, explain: "Sequence connectivity".

(Figure 3, 4) Could the orientation of the particle/train in the context of the cilium be added (plus/minus end of axoneme, the relative position of the membrane, and doublets)?

"Furthermore, the comparison of these ensembles would allow us to determine conserved domain interactions from our data (Figure S5)."

Should be Figure S6?

---

## [Author Response]

Essential revisions:The reviewers are concerned that there are very large differences between this model and the much higher resolution experimentally derived structures. To help you to re-evaluate your methodology and conclusions they suggest:1. Need documentation of the preparation used (SEC profile, SDS-PAGE, etc.) before and after X-linking. One reviewer thinks it is likely that IFT-A will no longer form a chain, so they are less concerned about A-to-A interactions in the XL experiments.

We now include the requested documentation, including confirmation of the monomeric nature of IFT-A in our samples by size exclusion chromatography (in the new Figure 1—figure supplement 2). Our revisions are discussed in full below in the responses to Reviewer #1.

2. Need to perform integrative modeling using sub-complexes predicted by α fold to get a better-corrected model. The fact that the model in part deviates from data presented in other preprint studies is concerning.

This was an excellent suggestion (and a well-deserved nudge) from the Reviewers, and we have now substantially refined the model using this strategy (as detailed below). We see better concordance between our model and the subtomogram average, which we now use exclusively to construct a polymeric IFT-A train. The improved model shows excellent agreement with the two contemporaneous preprints from Hesketh *et al.* and Lacey *et al.* This revision of the manuscript focuses entirely on the refined, higher confidence model.

3. Need to address deeply the similarities/differences of the cryo-ET map of Lacey et al. and other IFT-A models.

A direct comparison is not entirely trivial, as we were the only group to deposit our atomic coordinates with our preprint, but we now compare to figure panels in the preprints. While our initial monomer model was broadly concordant with the other monomer structures, especially in the IFT139/121/122/43 region of IFT-A, there were substantial disagreements with the placement of the model into the tomogram and the organization of IFT140/144. Following the refinement of our model to address point 2 above, there is now substantial agreement, particularly with the polymeric IFT-A form. We present these new comparisons on p. 10 and in the new Figure 4—figure supplement 2.

Reviewer #1 (Recommendations for the authors):1) From personal experience I know how difficult it can be to fit multiple subunits into low-resolution maps (manually or automatically) as completely different solutions often provide equally good fits to the map. This exercise becomes increasingly easier the fewer and larger the sub-complexes are. In the current study, the AlphaFold multimer would be an excellent resource to model dimeric and trimeric IFT-A sub-complexes, which could then be used as a starting point for the structural modeling of the complete IFT-A complex.

This proved to be an excellent recommendation. We revisited the modeling to compute AlphaFold-Multimer structures for all pairs of IFT-A proteins. Importantly, we only observed AlphaFold evidence for the interactions for those pairs for which we measured chemical cross-links, indicating a high level of agreement between our cross-linking data and AlphaFold. (We have a recent preprint comparing more than a thousand ciliary cross-links with AlphaFold structures and can confirm this high concordance more generally; see doi:10.1101/2022.08.25.505345). We then segmented the AlphaFold multimer structures to identify high-confidence protein interaction interfaces and folded domains, and used these as starting models for our full integrative molecular modeling pipeline as before. Because we confirmed the monomeric nature of our IFT-A particles (as discussed above), we used only AlphaFold and chemical cross-linking data as modeling constraints, withholding the 23 Å tomogram for subsequent analysis of the polymeric IFT-A structure. This process produced a model that was substantially more concordant with the preprint of Hesketh *et al.*, and importantly also suggested conformations of the IFT140/144 tails that were in better agreement with the chemical cross-linking datasets.

Reviewer #2 (Recommendations for the authors):I wonder if the Report format is optimal for this paper or if the traditional separation of results and discussion would be more appropriate.

Thank you for the suggestion. We now separate the Results and Discussion sections.

I also think the manuscript could be strengthened by including some documentation (silver stained gel or general MS) showing the IFT-A raw material used for the cross-linking experiments (or a similar prep), which is described only as "highly enriched endogenous IFT-A complexes". While the data show that the approach worked it would nevertheless be informative to know the relative enrichment of IFTA, which will inform on the robustness of the method and could be also helpful to apply the approach developed here to other particles.

We now include a new supplementary figure (Figure 1—figure supplement 2) documenting the biochemical fractions analyzed, including their enrichment for IFT-A and the measured molecular weight of the IFT-A particles, indicating they were monomeric. Using mass spectrometry, we estimate that IFT-A and IFT-B each comprised approximately 3% of the cross-linked sample, as we now note on p. 12.

Also, explanatory figure legends need to be provided for the supplementary figures.

Corrected. All supplementary figures now have legends.

"Accordingly, we observed 86% agreement between the human model and the Tetrahymena XL/MS data."Explain "86% agreement". Does it mean that the X-linked residues in Tetrahymena are predicted to be 35 nm or less apart in human IFT proteins by AlphaFold2?

This interpretation is correct, as we now clarify on p. 4 (and note that the distance is in units of Angstroms, not nm).

"9 + 2 microtubule axoneme, a microtubule-based cytoskeletal structure".

Corrected.

"followed by the movement of products back to the cell body by dynein motors"Not sure what is actually produced in cilia, products are potentially misleading.

Corrected.

"although some cargoes also move by diffusion (17,18)."Some "ciliary proteins" might be a better term as "cargo" implies transport by a motor-based process. Add Harris et al. MBoC to references for diffusion.

Corrected and cited.

"inter-molecular" and "intermolecular" are both used.

Corrected.

"We then selected the cluster that best agreed with the chemical cross-linking data with multiple runs converging on similar arrangements of the IFT-A complex, indicating that the model arrived independently from different starting configurations (see Figure S3, and Methods)."The sentence is not clear to me. I wonder if it would make sense to show some more deviant models/clusters with scores (how many x-links still fit within 3.5 nm?)

The modeling section has now been revised to reflect the new scheme used for our refined model. To address the latter comment, we also now include comparisons of starting and ending model configurations in the new Figure 2—figure supplement 3 and Figure 2—figure supplement 4, with histograms of crosslink distances illustrating the improvements seen in the final model.

Legend Figure 1B: Bar diagrams.

We have now expanded this legend for clarity.

Figure 4, legend: "For comparison, two models, each derived from half the data, exhibited a cross-correlation coefficient of 0.90."Meaning what? If the particle is cut in half, each half fits better into the outline based on cryo-EM.

We have now rewritten this section for improved clarity (now on pp. 5-6).

Legend Figure 5: I think the legend should be extended explaining the color code for the different species. The Roman numbered subpanels are explained in the text but a shorter version should be added to the legend.

Corrected.

P4, explain: "Sequence connectivity".

Corrected.

(Figure 3, 4) Could the orientation of the particle/train in the context of the cilium be added (plus/minus end of axoneme, the relative position of the membrane, and doublets)?

Corrected.

"Furthermore, the comparison of these ensembles would allow us to determine conserved domain interactions from our data (Figure S5)."Should be Figure S6?

Corrected.